# Prospects for Precise Measurements with Echo Atom Interferometry

**Brynle Barrett [1,2], Adam Carew [1], Hermina C. Beica [1,*], Andrejs Vorozcovs [1], Alexander Pouliot [1] and A. Kumarakrishnan [1]**

[1] Department of Physics & Astronomy, York University, 4700 Keele, Toronto ON M3J 1P3, Canada; brynle.barrett@institutoptique.fr (B.B.); accarew@gmail.com (A.C.); andrew.vorozcovs@gmail.com (A.V.); alexpouliot@live.com (A.P.); akumar@yorku.ca (A.K.)

[2] Institut d'Optique d'Aquitaine, rue François Mitterand, Talence 33400, France

[*] Correspondence: hermina@yorku.ca

**Abstract:** Echo atom interferometers have emerged as interesting alternatives to Raman interferometers for the realization of precise measurements of the gravitational acceleration $g$ and the determination of the atomic fine structure through measurements of the atomic recoil frequency $\omega_q$. Here we review the development of different configurations of echo interferometers that are best suited to achieve these goals. We describe experiments that utilize near-resonant excitation of laser-cooled rubidium atoms by a sequence of standing wave pulses to measure $\omega_q$ with a statistical uncertainty of 37 parts per billion (ppb) on a time scale of ∼50 ms and $g$ with a statistical precision of 75 ppb. Related coherent transient techniques that have achieved the most statistically precise measurements of atomic g-factor ratios are also outlined. We discuss the reduction of prominent systematic effects in these experiments using off-resonant excitation by low-cost, high-power lasers.

**Keywords:** atom interferometry; metrology; laser-cooling and trapping

## 1. Introduction

Over the past few decades, there has been sustained interest in using the exquisite sensitivity of atom interferometric techniques to gain a precise knowledge of the fundamental forces that govern our universe through an improved understanding of light-matter interactions. Among pioneering advances in this field include the diffraction of atomic beams using standing wave light fields [1,2] and micro-scale material beam splitters [3,4], and sensitive measurements of the index of refraction of an atomic gas [5]. The development of atom interferometers (AIs) to measure fundamental constants [6] and inertial effects [7,8] using laser-cooled atoms showed the potential of AIs for realizing precise studies of fundamental physics, and for industrial applications such as oil and mineral prospecting or inertial navigation. During the last 25 years, there has been steady progress toward developing AIs and coherent transient techniques [9–11] for measurements of fundamental constants such as $\alpha$ (or the ratio of Planck's constant to the mass of the test atom $h/M$) [12–16], the gravitational constant $G$ [17–20], studies of inertial effects such as gravitational acceleration [8,21,22], gravity gradients [23–25] and rotations [7,26–29], sensing magnetic gradients [30–33] and for more sensitive tests of the equivalence principle [34–42] and general relativity [43,44].

Most of the progress in both short-term and long-term sensitivity has been achieved using Raman interferometers [8,22,45–48]. This AI relies on optical velocity-sensitive two-photon Raman transitions between two long-lived hyperfine ground states in alkali atoms, such as the $|F = 1\rangle$ and

$|F = 2\rangle$ states in $^{87}$Rb. Raman AIs were the first implementation of state-labeled interferometer [45], where the coherent exchange of photon momentum between the atoms and the optical fields is associated with a change in the internal atomic state. Hence, all the information regarding the interference between atoms is stored in the relative population between states $|F = 1\rangle$ and $|F = 2\rangle$.

Despite the well-developed nature of Raman AIs, a number of alternate interferometer configurations have been developed [49–53], particularly for measurements of gravity. In this article, we report on recent developments and techniques for precision measurements using a unique, single-state echo interferometer [54,55]. This AI requires only a single excitation frequency, and does not require velocity selection. Recently, this configuration has achieved several milestones, including an extension of the timescale to the transit time limit (~250 ms for experimental conditions) [32], as well as significant improvements in statistical precision relating to measurements of the atomic recoil frequency (related to $h/M$) [56,57] and the acceleration due to gravity [58]. In what follows, we review recent progress on both atomic recoil (Section 2) and gravity (Section 3) measurements. In these sections, we also discuss various methods of reducing or eliminating the dominant systematic effects which are currently limiting the measurements. In Section 4, we review related coherent transient techniques [59,60] that have demonstrated precise measurements of atomic g-factor ratios. Finally, in Section 5 we describe the development of a new class of auto-locked semiconductor diode lasers operating at 780 nm and 633 nm [61]. These low-cost, high-power laser sources exhibit impressive long-term lock stability that will be implemented in future generations of the experiments described in Sections 2 and 3. Finally, we conclude with some perspectives in Section 6.

In the following subsections, we compare the operating principles of a Raman interferometer with those of a single-state grating-echo AI, including a brief theoretical description of the two types.

## 1.1. Description of Raman-Type Interferometers

As previously mentioned, Raman-transition-based interferometers rely on coherently transferring atoms between internal states of the atom. To make these transitions, two counter-propagating optical fields are used, one at frequency $\omega_1$ with wavevector $\boldsymbol{k}_1$, and the other at $\omega_2$ with $-\boldsymbol{k}_2$. These Raman fields satisfy the resonance condition $\omega_1 - \omega_2 = \omega_{\mathrm{HF}} + \boldsymbol{k}_{\mathrm{eff}} \cdot \boldsymbol{v} + \omega_{k_{\mathrm{eff}}}$ for making two-photon transitions between ground states $|F = 1, \boldsymbol{p}\rangle$ and $|F = 2, \boldsymbol{p} + \hbar\boldsymbol{k}_{\mathrm{eff}}\rangle$. Here, $\omega_{\mathrm{HF}}$ is the hyperfine splitting between $|F = 1\rangle$ and $|F = 2\rangle$, $\boldsymbol{p} = M\boldsymbol{v}$ is the initial momentum of the atom, $\boldsymbol{k}_{\mathrm{eff}} = \boldsymbol{k}_1 + \boldsymbol{k}_2$ is the effective wavevector of the counter-propagating Raman fields, and $\omega_{k_{\mathrm{eff}}} = \hbar k_{\mathrm{eff}}^2/2M$ is the atomic recoil frequency associated with the Raman transition.

The most basic and widely used type of Raman interferometer is the Mach-Zehnder configuration [8] which consists of a $\pi/2 - \pi - \pi/2$ sequence of pulses separated by a time $T$, as shown in Figure 1a. The first $\pi/2$-pulse acts as a beam-splitter that creates a 50/50 superposition of the states $|1, \boldsymbol{p}\rangle$ and $|2, \boldsymbol{p} + \hbar\boldsymbol{k}_{\mathrm{eff}}\rangle$. The atoms then travel along two spatially separated pathways. The $\pi$-pulse at $t = T$ acts as a mirror, exchanging the population between the two states and redirecting the wavepackets associated with each trajectory back toward one another. The final $\pi/2$-pulse at $t = 2T$ recombines the wavepackets by "closing" the interferometer pathways and producing the interference. The two output ports of the interferometer correspond to the relative populations in each state, for instance $N_1/(N_1 + N_2)$, where $N_F$ represents the number of atoms in state $|F\rangle$. These populations are usually measured via resonant fluorescence, where many photons can be scattered per atom. Since the Raman interferometer excites only two pathways, the fringe pattern follows a simple sinusoidal function

$$\frac{N_{1,2}}{N_1 + N_2} \equiv P_{1,2} = \frac{1}{2}\Big(1 \pm \mathcal{C}\,\cos\Delta\Phi_{\mathrm{tot}}\Big), \tag{1}$$

where $P_{1,2}$ is the probability of finding the atom in either state at the output of the interferometer, $\mathcal{C}$ is the contrast of the interference fringes, and $\Delta\Phi_{\mathrm{tot}}$ is the total interferometer phase difference. The key idea of a Raman interferometer is that the population between internal states oscillates as a function

of the phase difference between interfering pathways. In general, this phase consists of three main contributions [62]

$$\Delta\Phi_{\text{tot}} = \Delta\phi_{\text{prop}} + \Delta\phi_{\text{sep}} + \Delta\phi_{\text{las}}, \tag{2}$$

where $\Delta\phi_{\text{prop}} = \oint \mathcal{L}(\boldsymbol{r}, \boldsymbol{v})\mathrm{d}t/\hbar$ is the propagation phase corresponding to the difference of classical action (integral of the lagrangian $\mathcal{L}$) along the upper and lower pathways, $\Delta\phi_{\text{sep}} = \boldsymbol{p} \cdot \Delta\boldsymbol{r}/\hbar$ is a phase associated with a spatial separation $\Delta\boldsymbol{r}$ between the wavepackets during the final $\pi/2$-pulse, and $\Delta\phi_{\text{las}}$ is due to the Raman laser phase imprinted on the atoms during each pulse, which is given by

$$\Delta\phi_{\text{las}} = \boldsymbol{k}_{\text{eff}} \cdot \big(\boldsymbol{r}_{\text{c}}(0) - 2\boldsymbol{r}_{\text{c}}(T) + \boldsymbol{r}_{\text{c}}(2T)\big) + \varphi_1 - 2\varphi_2 + \varphi_3. \tag{3}$$

Here, $\boldsymbol{r}_{\text{c}}(t)$ represents the center-of-mass trajectory of the atom and $\varphi_j$ is the phase difference between the Raman fields at the $j^{\text{th}}$ pulse.

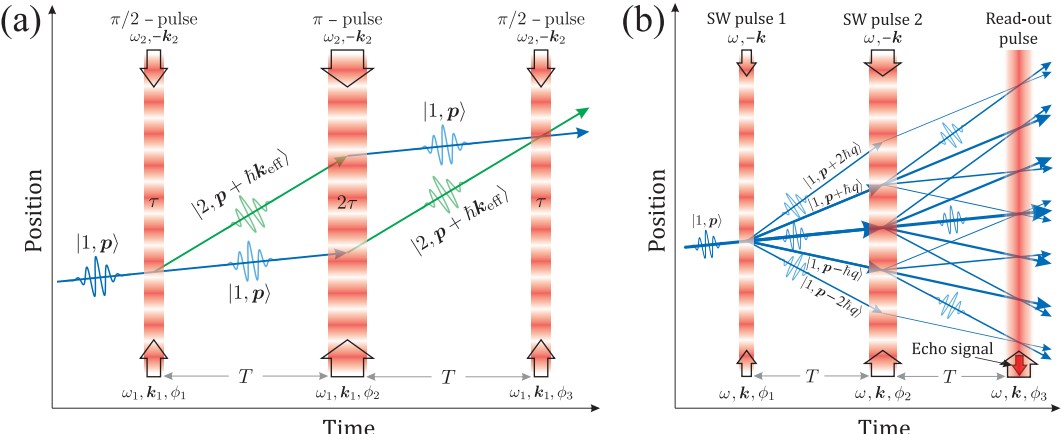

**Figure 1.** Raman and echo-type atom interferometer schemes. (**a**) Three-pulse Mach-Zehnder configuration of a Raman interferometer for measuring inertial effects such as gravity. Each light pulse is composed of two counter-propagating beams of frequencies $\omega_1$ and $\omega_2$ that induced velocity-sensitive Raman transitions between states $|1, \boldsymbol{p}\rangle$ (blue lines) and $|2, \boldsymbol{p} + \hbar\boldsymbol{k}_{\text{eff}}\rangle$ (green lines). The phase difference between these two beams during each pulse ($\varphi_j$) is imprinted on the wavepackets wherever a change of momentum takes place. In the absence of any accelerations, the final atomic populations at the output of the interferometer oscillate sinusoidally as a function of the total laser phase difference $\Delta\varphi = \varphi_1 - 2\varphi_2 + \varphi_3$. (**b**) Two-pulse configuration of the grating-echo AI, which is sensitive both to inertial effects and to the atomic recoil frequency. Each light pulse is composed of counter-propagating beams of the same frequency $\omega$ and polarization, which create a standing wave with phase $\varphi_j$. The two SW pulses, separated by a time $T$, diffract atoms in the same internal state $|1, \boldsymbol{p}\rangle$ into a superposition of momentum states $|1, \boldsymbol{p} + n\hbar\boldsymbol{q}\rangle$ separated by integer multiples of the two-photon momentum $\hbar\boldsymbol{q} = 2\hbar\boldsymbol{k}$. At time $t = 2T$, a subset of these momentum states interfere and create a spatially modulated density grating with period $\lambda/2$. A traveling-wave read-out pulse is applied at this time, and the back-scattered "echo" signal from the atoms is measured. The phase of this back-scattered light is proportional to the phase shift of the interference pattern due to gravity. Similarly, the back-scattered light intensity is modulated as a function of $T$ at the atomic recoil frequency.

Due to the limited bandwidth of two-photon Raman transitions given by the Rabi frequency $\Omega_{\text{eff}}/2\pi \simeq 10 - 100\,\text{kHz}$, and the desire to use only atoms in magnetically "insensitive" $m_F = 0$ states, a large percentage of atoms are lost during the sample preparation process. To give a typical example, a sample of $N \sim 10^8$ laser-cooled $^{87}$Rb atoms at a temperature of $\sim 5\,\mu\text{K}$ has an initial velocity spread of $\sigma_v \simeq 3\,\text{cm/s}$. After preparing the atoms in the $|F = 1, m_F = 0\rangle$ state using a sequence of near-resonant push beams and microwave $\pi$-pulses, typically 1/3 of the atoms remain. Using a velocity-selective

Raman pulse with $\Omega_{\text{eff}}/2\pi \simeq 10$ kHz transfers a narrow velocity class ($\sigma'_v = \Omega_{\text{eff}}/k_{\text{eff}} \simeq 0.4$ cm/s) of atoms to the $|F = 2, m_F = 0\rangle$ state, while the remaining atoms are removed—resulting in an 8-fold loss in atom number. Thus, in total, Raman AIs exhibit atom number loss factors on the order of $\sim 25$. In this example, approximately $N \sim 4 \times 10^6$ contribute to the interferometer. However, each atom can scatter several thousand photons during the resonant detection pulses, thereby ensuring an adequate signal-to-noise ratio. A measure of a Raman interferometer's sensitivity is the so-called shot-noise or quantum-projection-noise limit [63], where the Poissonian fluctuations of atomic state measurements limit the minimum uncertainty of individual phase measurements to $\delta\phi_{\text{shot}} = 1/\sqrt{N}$. Shot-noise limits of $<1$ mrad are considered to be state-of-the-art [64,65].

For the simple example of the Earth's gravitational potential, where $\mathcal{L} = Mv^2/2 - Mgz$ and $r_c(t) = \left(z_0 + (v_0 + \frac{\hbar k_{\text{eff}}}{2M})t - \frac{1}{2}gt^2\right)\hat{z}$, it is straightforward to show that the first two phase terms in Equation (2) vanish—leaving only the laser phase. Hence, if the Raman beams are aligned along $-\hat{z}$ such that $\boldsymbol{k}_{\text{eff}} = -k_{\text{eff}}\hat{z}$, the total phase shift is

$$\Delta\Phi_{\text{tot}} = k_{\text{eff}}gT^2 + \Delta\varphi, \tag{4}$$

with $\Delta\varphi = \varphi_1 - 2\varphi_2 + \varphi_3$, which is usually used as a control parameter in the experiment to scan the interference fringes—enabling a direct measurement of the gravitational acceleration. Equation (4) illustrates the strong sensitivity of atom interferometers to inertial effects such as gravity. Since the phase shift scales as $k_{\text{eff}}T^2$, with a modest interrogation time of $T \sim 50$ ms and $k_{\text{eff}} \sim 1.6 \times 10^7$ rad/m for light at wavelength $\lambda = 780$ nm, the acceleration due to gravity induces a phase shift of $\Delta\Phi_{\text{tot}} \simeq 4 \times 10^5$ rad. Hence, with a phase uncertainty of 1 mrad, the single-shot sensitivity of the interferometer is $\sim 3 \times 10^{-9} g$. State-of-the-art cold-atom gravimeters have demonstrated precisions of 0.2 ppb after 1000 s of integration [47].

For measurements of the atomic recoil frequency with a Raman interferometer, typically the Ramsey-Bordé configuration is used [14–16,45]. In this case, the central $\pi$-pulse is replaced with two $\pi/2$-pulses separated by a time $T'$, which has the effect of spatially separating the two output ports of the interferometer. The phase difference between interfering pathways is then $\Delta\Phi_{\text{tot}}^{\pm} = 2\omega_{k_{\text{eff}}}T \pm k_{\text{eff}}gT(T + T')$, where the $\pm$ corresponds to the upper and lower output ports, respectively [66]. The phase shift due to gravity can be rejected by operating the interferometer in a conjugate mode where the upper and lower ports are detected simultaneously—yielding the sum of the two phases $\Delta\Phi_{\text{tot}} = 4\omega_{k_{\text{eff}}}T$ [67]. To increase the sensitivity to the recoil frequency $\omega_{k_{\text{eff}}}$, large momentum transfer beam-splitters such as high-order Bragg transitions, have been used in place of two-photon Raman transitions [68–70]. This has the effect of replacing the effective wave with $nk_{\text{eff}}$ in the equations above, thus the recoil phase of the conjugate Ramsey-Bordé interferometers becomes $4n^2\omega_{k_{\text{eff}}}T$. Bloch oscillations have also been used to increase the common momentum transfer to the atoms between the two pairs of $\pi/2$-pulses [14,15]. In this case, the recoil frequency is measured from the phase shift between the two Ramsey fringe patterns created with the first and last pairs of $\pi/2$-pulses [6]. This phase shift is proportional to the number of photon momenta transferred to the atoms by the Bloch oscillations, where transfers as large as 1600 photon momenta have been demonstrated [14,71]. Presently, the state-of-the-art in terms of precision for a measurement of $h/M$ is $1.2 \times 10^{-9}$ [15].

### 1.2. Description of Grating-Echo-Type Interferometers

The grating-echo AI is a single-state Talbot-Lau interferometer [72–74], the principles of which can be understood on the basis of a plane-wave description of the two-pulse scheme shown in Figure 1b [32,54,55,75–77]. This AI relies on matter-wave interference produced by Kapitza-Dirac scattering of atoms by short, off-resonant standing wave (SW) pulses. Typically, the interferometer uses a sub-Doppler-cooled atomic sample with a momentum spread $\sigma_p \gg \hbar k$, initially prepared in a single hyperfine ground state $|F\rangle$. Two SW pulses are applied to the sample separated by a time $T$.

The first pulse excites a superposition of momentum states separated by integer multiples of $\hbar q$. The second excitation pulse further diffracts the momentum states, causing certain trajectories to interfere in the vicinity of $t = 2T$, henceforth referred to as the "echo" time. This interference creates a spatial modulation in the atomic probability density with a phase that is proportional to inertial effects (i.e., $\phi_a = q \cdot a T^2$ due to an acceleration $a$), and a contrast that is temporally modulated at a harmonic of the two-photon recoil frequency $\omega_q = \hbar q^2 / 2M$. To measure the properties of this interference, a unique optical detection scheme is used—a traveling wave read-out pulse is applied the sample in the vicinity of the echo time when the density modulation is strongest. A certain spatial harmonic of this modulation satisfies the Bragg condition for scattering light in the backward direction (i.e., the harmonic with period $\lambda/2$) as shown in Figure 1b. This back-scattered "echo signal" carries both the phase and contrast information about the atomic interference between certain classes of trajectories—namely those whose momenta differ by $\hbar q$ at $t = 2T$.

A common feature of the echo AI experiments described in this article is that the contrast of the density modulation (grating) is small, due to the relatively small atom-field coupling strength and the short pulse durations of the SW pulses. Consequently, the experiments are limited by the strength of the signal, which is defined by the reflectivity of the grating ($\approx 0.2\%$). So although echo experiments do not experience the appreciable atom loss characteristic of Raman AIs, they require large atom numbers and high-contrast gratings to achieve appreciable signal strengths.

To understand how the atomic density grating comes about, we consider two overlapping momentum states $|F, n\hbar q\rangle$ and $|F, n'\hbar q\rangle$ labelled by integers $n$ and $n'$. In comparison to Raman and Bragg interferometers [8,78], no atom optical "combiner" pulse is required to produce interference between two wavepacket trajectories since the momenta are in the same hyperfine ground state $|F\rangle$. Spatial overlap is the only condition required to create an interference pattern, which can be described by

$$\langle F, n'\hbar q \,|\, F, n\hbar q \rangle \sim e^{i(n-n')q\cdot r} e^{-i\Delta\phi_{n,n'}}. \tag{5}$$

Here, $\Delta\phi_{n,n'}$ is the phase difference between the wavepackets, which has contributions from the Doppler shift, atomic recoil and the SW laser phase, as we discuss below. If the integers $n$ and $n'$ satisfy $n' = n \pm 1$, then the real part of this interference is $\sim\cos(q \cdot r + \Delta\phi_{n,n\pm1})$. The density distribution follows this simple sinusoidal pattern, which exhibits a period of $\lambda/2$ and hence satisfies the Bragg scattering condition for detection. In reality, the pair of SW pulses excite multiple interfering trajectories—each contributing its own spatial harmonic to the density distribution. To account for this multi-path interference, one must sum over all possible trajectories to arrive at the correct interference pattern. Although this can lead to extremely complex periodic structures in the atomic density [55,75], a simplification that can always be made is the fact that the read-out light will only scatter from the $q$-Fourier component of this structure.

We now give a detailed description of the plane-wave theory of grating-echo formation. Initially, the atomic wavefunction is in a hyperfine ground state labeled by total angular momentum $F$ with momentum $p$, thus the wave function before the first SW pulse can be written as

$$|\psi_0(p, t)\rangle = |F, p\rangle \, e^{-i(\omega_0 + \omega_p)t}, \tag{6}$$

where $\hbar\omega_0$ is the internal energy of the ground state, and $\omega_p = p^2/2M\hbar$ is the frequency associated with the initial kinetic energy of the atom. Since the phase term $e^{-i(\omega_0 + \omega_p)t}$ is common to all diffracted momentum states, it is unimportant for interference and we henceforth ignore it. In the short pulse duration (i.e., Raman-Nath) regime, the interaction with the off-resonant SW pulse simply modulates the phase of the wavefunction as $e^{-i\Omega_{\text{eff}}\tau_j \cos(q\cdot r + \varphi_j)} |\psi_0\rangle$, where $\Omega_{\text{eff}}$ is the effective Rabi frequency of the light, $\tau_j$ is the duration of the $j^{\text{th}}$ pulse and $\varphi_j$ is the phase of the standing wave (usually

defined by the location of the node created by a retro-reflecting mirror). We can use the Jacobi-Anger expansion [79] to describe this modulation in a more convenient form

$$e^{-i\Omega_{\text{eff}}\tau_j \cos(\boldsymbol{q}\cdot\boldsymbol{r}+\varphi_j)} = \sum_{n=-\infty}^{\infty} (-i)^n J_n(\Omega_{\text{eff}}\tau_j) e^{in(\boldsymbol{q}\cdot\boldsymbol{r}+\varphi_j)}, \tag{7}$$

where $J_n(x)$ is the $n^{\text{th}}$ order Bessel function of the first kind. Thus, after the first standing wave pulse applied at time $t = t_1$, the wavefunction can be shown to be

$$|\psi_1(\boldsymbol{p},t)\rangle = \sum_n A_n |F, \boldsymbol{p} + n\hbar\boldsymbol{q}\rangle \, e^{in\varphi_1} e^{-in\boldsymbol{q}\cdot\boldsymbol{v}(t-t_1)} e^{-in^2\omega_q(t-t_1)}, \tag{8}$$

where $A_n = (-i)^n J_n(\Omega_{\text{eff}}\tau_1)$ and $\boldsymbol{v} = \boldsymbol{p}/M$ is the initial center-of-mass velocity of the atom. Here, we have used the fact that $|F, \boldsymbol{p}\rangle \, e^{in\boldsymbol{q}\cdot\boldsymbol{r}} = |F, \boldsymbol{p} + n\hbar\boldsymbol{q}\rangle$. After the second standing wave pulse at time $t = t_2 = t_1 + T_{21}$, the wavefunction is given by

$$|\psi_2(\boldsymbol{p},t)\rangle = \sum_{n,m} A_n B_m |F, \boldsymbol{p} + (n+m)\hbar\boldsymbol{q}\rangle \, e^{i(n\varphi_1 + m\varphi_2)}$$
$$\times e^{-i\boldsymbol{q}\cdot\boldsymbol{v}[nT_{21}+(n+m)(t-t_2)]} e^{-i\omega_q[n^2 T_{21}+(n+m)^2(t-t_2)]}, \tag{9}$$

where $T_{21} = t_2 - t_1$ and $B_m = (-i)^m J_m(\Omega_{\text{eff}}\tau_2)$. To find the interference pattern at time $t = t_1 + 2T_{21}$, we compute the atomic density distribution $\rho_2 = \langle\psi_2|\psi_2\rangle$

$$\rho_2(\boldsymbol{p},t) = \sum_{n,m,n',m'} A_n A_{n'}^* B_m B_{m'}^* \langle F, \boldsymbol{p} + (n'+m')\hbar\boldsymbol{q} | F, \boldsymbol{p} + (n+m)\hbar\boldsymbol{q}\rangle \, e^{i[(n-n')\varphi_1 + (m-m')\varphi_2]}$$
$$\times e^{-i\boldsymbol{q}\cdot\boldsymbol{v}[(n-n')T_{21}+(n+m-n'-m')(t-t_2)]} e^{-i\omega_q\{(n^2-n'^2)T_{21}+[(n+m)^2-(n'+m')^2](t-t_2)\}}. \tag{10}$$

This time-dependent expression, although complex, has a simple interpretation. Each term in the sum is composed of three factors: (i) the complex amplitude factor $A_n A_{n'}^* B_m B_{m'}^*$ which determines the relative strength of different interfering trajectories, (ii) the interference term $\langle F, \boldsymbol{p} + (n'+m')\hbar\boldsymbol{q} | F, \boldsymbol{p} + (n+m)\hbar\boldsymbol{q}\rangle \sim e^{-i(n+m-n'-m')\boldsymbol{q}\cdot\boldsymbol{r}}$, which produces a modulation in the atomic density with spatial harmonic $(n+m-n'-m')\boldsymbol{q}$, and (iii) a series of phase factors that modify the phase of the density modulation due to the laser interaction ($e^{i[(n-n')\varphi_1+(m-m')\varphi_2]}$), the Doppler shift ($e^{-i\phi_{\text{D}}(t)}$), and the atomic recoil ($e^{-i\phi_q(t)}$), where

$$\phi_{\text{D}}(t) = \boldsymbol{q}\cdot\boldsymbol{v}[(n-n')T_{21} + (n+m-n'-m')(t-t_2)], \tag{11a}$$
$$\phi_q(t) = \omega_q\{(n^2-n'^2)T_{21} + [(n+m)^2 - (n'+m')^2](t-t_2)\}. \tag{11b}$$

The set of integers $\{n, m, n', m'\}$ label the momentum (in units of $\hbar\boldsymbol{q}$) transferred to the atom by the SW pulses, and represent a particular pair of interfering trajectories in Figure 1b. For instance, the integer labels corresponding to the trapezoidal trajectories of the Mach-Zehnder geometry shown in Figure 1a are $\{n, m, n', m'\} = \{1, -1, 0, 1\}$ (Here, we interpret the unprimed integers $\{n, m\}$ as the momenta transferred along the upper pathway, while the primed integers $\{n', m'\}$ correspond to the lower pathway of any two trajectories.). It follows that the interference between these trajectories produces a density modulation with a period of $2\pi/|n+m-n'-m'|q = 2\pi/q = \lambda/2$, which is the ideal period for back-scattering the electric field of the read-out pulse at wavelength $\lambda$.

Since the velocity distribution of the sample is assumed to be broad compared to the scale of momentum transfer ($\sigma_p \gg \hbar k$), the macroscopic density grating produced in the sample is found by averaging the single-atom probability density (10) over this distribution of velocities. However, the velocity-dependent Doppler phase causes a strong dephasing effect on the grating at all times except certain "echo" times where this phase is zero [55,75,80]. One can show that these times must satisfy $t = t_1 + (1 - \delta_1/\delta_2)T_{21}$, where $\delta_1 = n - n'$ and $\delta_2 = n + m - n' - m'$. Here, we are concerned with

only the first echo time at $t_1 + 2T_{21}$, which implies a ratio of $\delta_1/\delta_2 = -1$. The back-scattering detection method constrains $\delta_2 = \pm 1$, thus we require $\delta_1 = \mp 1$ (i.e., $n' = n \pm 1$). Inserting this constraint into $\delta_2$ yields $m' = m \mp 2$. These two constraints define the class of trajectories that produce a macroscopic interference pattern at $t = t_1 + 2T_{21}$ which can back-scatter light at wavelength $\lambda$. This interference pattern, which represents only a subset of the total density modulation given by Equation (10), can be shown to be

$$
\begin{aligned}
\tilde{\rho}_2(\boldsymbol{r}, t) \sim & \sum_{s=-1,1} e^{-is\boldsymbol{q}\cdot\boldsymbol{r}} e^{-is(\varphi_1 - 2\varphi_2)} e^{-is\boldsymbol{q}\cdot\boldsymbol{v}(t-t_1-2T_{21})} e^{is^2\omega_q(t-t_1)} \\
& \times \sum_n A_n A_{n+s}^* e^{-i2ns\omega_q(t-t_1-2T_{21})} \sum_m B_m B_{m-2s}^* e^{-i2ms\omega_q(t-t_1-T_{21})}.
\end{aligned}
\tag{12}
$$

We now average over the velocity distribution of the sample, which is assumed to be a Maxwell-Boltzmann distribution $N(\boldsymbol{v}) = \frac{1}{\pi^{3/2}\sigma_v^3} e^{-(v/\sigma_v)^2}$ with $e^{-1}$ radius $\sigma_v$

$$
\begin{aligned}
\langle \tilde{\rho}_2(\boldsymbol{r}, t) \rangle \sim & \; e^{-[q\sigma_v(t-t_1-2T_{21})/2]^2} \cos(\boldsymbol{q}\cdot\boldsymbol{r} + \varphi_1 - 2\varphi_2) \\
& \times J_1\big[2\Omega_{\text{eff}}\tau_1 \sin\big(\omega_q(t-t_1-2T_{21})\big)\big] J_2\big[2\Omega_{\text{eff}}\tau_2 \sin\big(\omega_q(t-t_1-T_{21})\big)\big].
\end{aligned}
\tag{13}
$$

Here, we have made use of the Bessel function identity $\sum_\alpha J_\alpha(x) J_{\alpha+\beta}(x) e^{i\alpha\phi} = (i)^\beta e^{-i\beta\phi} J_\beta(2x\sin\phi)$ [79] to simplify the sums over $n$ and $m$ in Equation (12). Two important features of the interference pattern are now clear. First, as a result of velocity dephasing, the grating only exhibits non-vanishing contrast for a timescale of $2/q\sigma_v \simeq 1\ \mu$s in the vicinity of the echo time. Second, the atomic recoil frequency, which initially appeared in the phase of the wavefunction, now affects only the *contrast* of the interference pattern. This feature of echo AIs alleviates the need for phase sensitivity in a recoil-sensitive experiment, since the effect can be measured in the back-scattered signal *intensity*. This type of AI has also been referred to as a "contrast" interferometer in the context of recoil measurements with ultra-cold atoms [81,82].

The final step is to compute from this macroscopic density the signal that is detected in the experiment by back-scattering the traveling wave read-out light. The physical mechanism that generates this light is elastic Rayleigh scattering from a spatial modulation of the sample's refractive index that satisfies the Bragg condition [83–85]. This coherent scattering process results from a phase-matching condition along the Bragg angle. Whereas the intensity of diffuse atomic scattering scales as the number of scatters $N$, here the intensity scales as $N^2$—a well-known feature of coherent Bragg scattering [84]. The drawback of this process is that, since it depends on a coherent superposition of momentum states, each atom scatters at most one photon before being projected into one of the two states. In comparison, the incoherent process of near-resonant fluorescence used in Raman interferometers allows one to scatter many photons per atom to increase the signal-to-noise ratio. This emphasizes the need for large atom numbers, low-sample temperatures, and high-contrast gratings to reach signal-to-noise ratios comparable with Raman AIs.

The macroscopic density grating described by Equation (13) acts as a linear reflector for light of wavelength $\lambda$ [85,86]. Thus, a traveling-wave read-out field of $E_{\text{RO}} e^{i(\boldsymbol{k}\cdot\boldsymbol{r} - \omega t + \varphi_3)}$ couples to the atomic grating and produces a back-scattered field given by

$$
E_{\text{BS}}(t) \sim r(t) E_{\text{RO}} e^{i(-\boldsymbol{k}\cdot\boldsymbol{r} - \omega t + \varphi_3)},
\tag{14}
$$

where $r(t)$ is a time-dependent reflection coefficient [85], which depends on the detuning of the read-out light, and the contrast of the density modulation at spatial frequency $q$. Hence, for a fixed detuning, the back-scattered field is proportional to the probability density given by Equation (13)

$$
\begin{aligned}
E_{\text{BS}}(t) \sim & \; E_{\text{RO}} e^{-i(\boldsymbol{k}\cdot\boldsymbol{r} + \omega t)} e^{i(\varphi_1 - 2\varphi_2 + \varphi_3)} e^{-[q\sigma_v(t-t_1-2T_{21})/2]^2} \\
& \times J_1\big[2\Omega_{\text{eff}}\tau_1 \sin\big(\omega_q(t-t_1-2T_{21})\big)\big] J_2\big[2\Omega_{\text{eff}}\tau_2 \sin\big(\omega_q(t-t_1-T_{21})\big)\big].
\end{aligned}
\tag{15}
$$

This back-scattered field contains all the information about the atomic interference between momentum states differing by $\hbar q$. For instance, the time-integrated power of the back-scattered light (referred to as the echo energy) is a measure of the contrast produced by this interference. Experiments utilizing the two-pulse AI, where the echo energy is measured as a function of $T_{21}$, are described in Refs. [32,54–57,76,77].

Similarly, the effect of gravity on the echo AI is to shift the phase of the atomic grating, which in turn causes a phase shift on the back-scattered electric field. In the same spirit as described in Section 1.1, the phase shift due to gravity can be computed solely by considering the interaction with the lasers. From Equation (15), the laser phase has the same form as for the Raman interferometer: $\Delta\phi_{\text{las}} = \Delta\varphi \equiv \varphi_1 - 2\varphi_2 + \varphi_3$. By replacing the each $\varphi_j$ with $\varphi_j + \boldsymbol{q} \cdot \boldsymbol{r}_{\text{c}}(t_j)$, where $\boldsymbol{r}_{\text{c}}(t) = \boldsymbol{r}_0 + \boldsymbol{v}_0 t + \frac{1}{2}\boldsymbol{g}t^2$ is the center-of-mass trajectory of the atom under gravity, we find at $t = t_1 + 2T_{21}$

$$\Delta\phi_{\text{las}} = \boldsymbol{q} \cdot \boldsymbol{g}T_{21}^2 + \Delta\varphi. \tag{16}$$

Echo AI experiments that have demonstrated sensitivity to this gravitational phase shift are described in Refs. [30,54,55,58,87].

## 2. Measurements of Atomic Recoil Frequency

### 2.1. Introduction

There is an ongoing, international effort to develop precise, independent techniques for measuring the atomic fine structure constant, $\alpha$—a dimensionless parameter that quantifies the strength of the electromagnetic force which lies at the heart of light-matter interactions. These measurements can be used to stringently test the theory of quantum electrodynamics (QED). Historically, two types of determinations of $\alpha$ have been carried out: (i) those that use other precisely measured quantities to determine $\alpha$ through challenging QED calculations [88,89], and (ii) those that are independent of QED, and depend on only the quantities appearing in the definition $\alpha \equiv e^2/2\epsilon_0 hc$, where $e$ is the elementary charge, $\epsilon_0$ is the vacuum permittivity, $h$ is Planck's constant and $c$ is the speed of light. Some examples of $\alpha$ determinations that require QED are the measurements of the anomalous magnetic moment of the electron (precise to 0.37 ppb) [90], and the fine structure intervals of helium (precise to 5 ppb) [91]. The most precise examples of QED-independent determinations are those based on measurements of the von Klitzing constant, $R_{\text{K}} = h/e^2$, using the quantum-Hall effect [92,93], and the ratio $h/M$ using (i) Bloch oscillations in cold atoms [13,94] and (ii) atom interferometric techniques [6,12,14,15]. Within these examples, atom interferometry has emerged as a powerful tool because of its inherently high sensitivity to $h/M$, which can be related to $\alpha$ according to

$$\alpha^2 = \frac{2R_\infty}{c}\frac{h}{m_e} = \frac{2R_\infty}{c}\left(\frac{M}{m_e}\right)\left(\frac{h}{M}\right). \tag{17}$$

Here, $R_\infty$ is the Rydberg constant, $m_e$ is the electron mass, and $M$ is the mass of the test atom. Since $R_\infty$ is known to 6 parts in $10^{12}$, and the mass ratio $M/m_e$ is typically known to a few parts in $10^{10}$ [95], the quantity that limits the precision of a determination of $\alpha$ using Equation (17) is the ratio $h/M$. The most precise measurement of this ratio was recently carried out with $^{87}$Rb, where $h/M$ was determined to 1.2 ppb after 15 hours of data acquisition [15]. The corresponding determination of $\alpha$ was precise to 0.66 ppb. Other interferometric techniques that have demonstrated high sensitivity to $h/M$ include Refs. [16,81,82,96–99].

In this section, we describe recent improvements in measurements of the atomic recoil frequency $\omega_q = \hbar q^2/2M$ using echo AIs [54,55,57]. The appeal of this type of AI lies in it's reduced sensitivity to common systematic effects, such as phase shifts due to the AC Stark or Zeeman effects, since it involves only a single internal state. In addition, since echo AIs rely on short-duration standing-wave pulses, only a single laser is required, and the large bandwidth of these pulses alleviates the need

for velocity selection. Finally, as mentioned in Section 1.2, the signature of atomic recoil affects only the *contrast* of the interference pattern and is insensitive to low-frequency phase noise of the standing wave. Thus, in comparison to Raman interferometers, a phase-stable apparatus is not required to make high-precision measurements of $\omega_q$.

We have developed a "modified" three-pulse echo AI (This configuration is distinct from the "stimulated" three-pulse AI used for measurements of $g$ that we present in Section 3.5.3.) which exhibits increased sensitivity to atomic recoil compared to the aforementioned two-pulse configuration [57]. This modified geometry has been described in previous work using the formalism of coherence functions [100] and a full quantum-mechanical treatment that accurately describes the fringe shape [54,56,100]. In the same articles, we also discussed connections to $\delta$-kicked rotors and quantum chaos, as well as scaling laws that apply to excitation with multiple SW pulses that have been used in other work [101,102].

## 2.2. Description of the Modified Three-Pulse AI

For the modified three-pulse AI described in this section, an additional SW pulse is applied between the first two pulses at $t = \delta T$, as shown in Figure 2b [54,56,100–102]. This pulse has the effect of diffracting the atom into higher-order momentum states that contribute additional harmonics of $\omega_q$ to the temporal modulation of the macroscopic grating contrast. Intuitively, the third SW pulse acts as a phase mask analogous to the function of a multi-slit pattern in classical optics. More specifically, this pulse shifts the phase of the momentum states by $\eta \omega_q \delta T$, where $\eta$ is an integer that depends on the particular pathways that lead to interference at $t = 2T$. Hence, varying the time of this pulse $\delta T$ is analogous to moving the slit pattern along the propagation axis of light—yielding periodic revivals of the contrast of the interference pattern. An example of a pair of low-order interfering trajectories created by this interferometer is shown in Figure 2b (We emphasize that only a small subset of the trajectories excited by the SW pulses will interfere at $t = 2T$ for an arbitrary third pulse time, $\delta T$ (i.e., trajectories which, when combined, exhibit a Doppler phase that is independent of $\delta T$). Specifically, the only momentum states contributing to the signal are those that differ by $\hbar q$ after SW3 *and* after SW2.). When one accounts for all possible trajectories, the resulting signal consists of a series of narrow fringes separated by the recoil period, $\tau_q = \pi/\omega_q$ (~32 $\mu$s for rubidium), as a result of the interference between all excited momentum states differing by $\hbar q$.

When all relevant trajectories are summed over, it can be shown [56,100] that the resulting echo energy is modulated by $J_0[2u_3 \sin(\omega_q \delta T)]^2$, provided the third pulse area $u_3 = \Omega_{\text{eff}} \tau_3 \lesssim 1$. Here, $J_0(x)$ is the zeroth-order Bessel function of the first kind, $\Omega_{\text{eff}}$ is the effective two-photon Rabi frequency, and $\tau_3$ is the third SW pulse duration. Figure 2c illustrates the predicted dependence of the echo energy—that is, the energy in the back-scattered electric field—as a function of $\delta T$. The sensitivity of this AI to $\omega_q$ scales inversely with the time scale $T$ over which the signal can be measured, and it scales in proportion to the width of the fringes. The advantage of using this AI over the two-pulse configuration is the ability to narrow the fringe width using the third pulse. Additionally, since $T$ is fixed, the same number of atoms remain in the excitation beams at the time of detection—avoiding a loss of signal with increasing $\delta T$ due to effects like the thermal expansion of the sample. The fringe width is effectively determined by the width of the excited momentum distribution. By increasing the proportion of high-order momentum states (and thus the proportion of high-order harmonics of $\omega_q$) that contribute to the signal, the fringes become more sharply defined. The excitation is controlled by the interaction strength and duration of the third SW pulse. It can be shown that for small pulse durations (i.e., $\tau_3 \ll (\Omega_{\text{eff}} \omega_q)^{-1/2}$) the full-width at half-maximum (FWHM) of the fringe scales as $1/u_3$, as illustrated in Figure 2c [54,56,100].

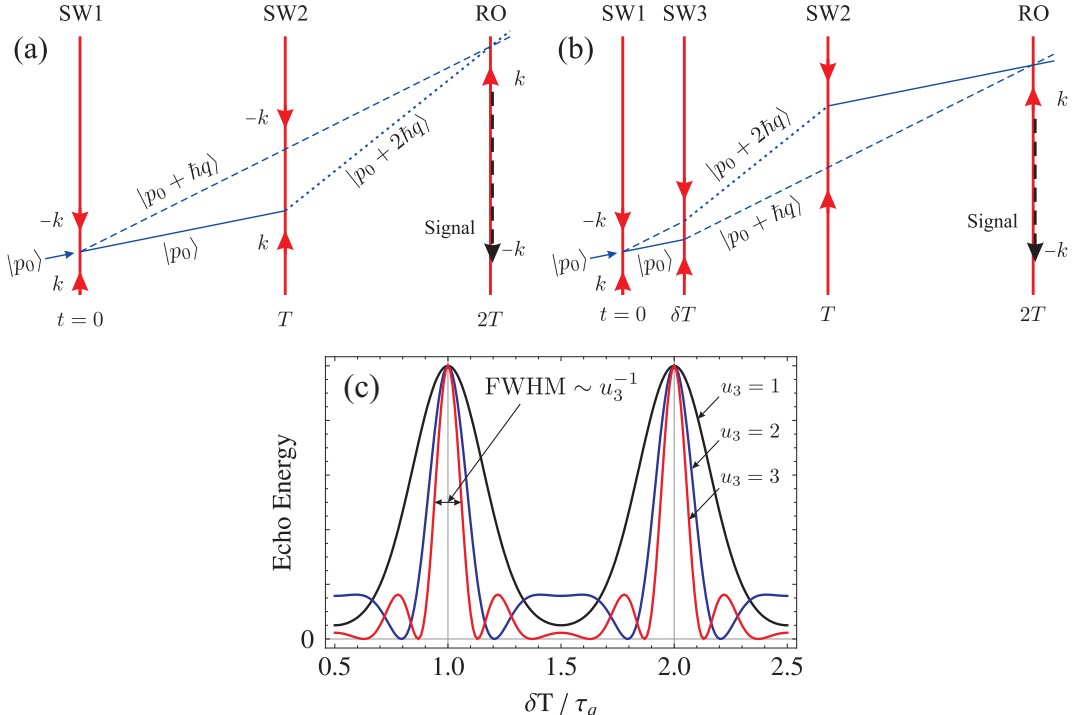

**Figure 2.** (**a**) An example of two, low-order trajectories that contribute to the two-pulse atom interferometer (AI) (SW$j = j^{\text{th}}$ SW pulse, RO = read-out pulse). Three momentum states are shown ($|p_0\rangle$, $|p_0 + \hbar q\rangle$, and $|p_0 + 2\hbar q\rangle$) corresponding to the solid, dashed, and dotted lines, respectively. The contrast of the interference fringe created at the echo time $t = 2T$ is modulated sinusoidally by a phase $2\omega_q T$. (**b**) Low-order trajectories for the modified three-pulse AI. Here, a third pulse is applied at $t = \delta T$ between SW1 and SW2 which further modulates the phase of the interference by $2\omega_q \delta T$. (**c**) Echo energy as a function of $\delta T/\tau_q$ for the modified AI. Line shapes are shown for three different pulse areas, $u_3 = \Omega_{\text{eff}}\tau_3$, to illustrate the effect of fringe narrowing that occurs for increasing interaction strength.

## 2.3. Experimental Setup

As described in Refs. [32,56,57], two major improvements to our experiment have enabled us to reach time scales of $T \simeq 60$ ms: (i) utilizing a non-magnetizable glass vacuum system, which reduced decoherence effects related to inhomogeneous *B*-fields [58] and improved the molasses cooling of the sample, and (ii) using large-diameter, chirped excitation beams, which increased the transit time of the atoms in the beam and compensated for the Doppler shift due to gravity (The non-uniform magnetic field produced by a stainless-steel vacuum chamber, and the gravity-induced Doppler shift limited previous experiments to $T \lesssim 10$ ms [76,100].).

The experiment utilizes a laser-cooled sample of rubidium typically containing $\sim 5 \times 10^9$ atoms at temperatures of $\mathcal{T} \lesssim 5$ $\mu$K. Either $^{85}$Rb or $^{87}$Rb atoms are loaded into a six-beam, vapor-loaded magneto-optical trap (MOT) in 250 ms. Prior to the AI experiment, the sample is prepared in the upper hyperfine atomic ground state ($5S_{1/2}$ $F = 3$ for $^{85}$Rb or $F = 2$ for $^{87}$Rb). The light for the AI is derived from a Ti:sapphire laser (linewidth $\sim$1 MHz) that is locked above the D2 cycling transition using Doppler-free saturated absorption spectroscopy. A network of acousto-optic modulators (AOMs) is used to generate the frequencies necessary for the AI excitation and the read-out beams. The read-out light is detuned to the blue of the cycling transition by $\Delta_{\text{RO}} = 40$ MHz, which optimizes the back-scattered signal intensity for our sample size and density [56]. The AI beams are detuned by $\Delta_{\text{AI}} = 220$ MHz, and a frequency chirp of $\delta(t) = gt/\lambda$ is added to (subtracted from) the downward- (upward-) traveling component of the SW pulses. This compensates the Doppler shift

induced by the falling atoms, and ensures that the beams remain resonant with the two-photon transition [32,56,57]. A "gate" AOM is used upstream of the AI AOMs as both a frequency shifter and a high-speed shutter to reduce the amount of stray light in the experiment. All RF sources and digital-delay generators used to define the pulse timing for the AI are externally referenced to a 10 MHz rubidium clock.

The AI beams are coupled into two AR-coated, single-mode optical fibers and aligned through the sample, as shown in Figure 3a. At the output of the fibers, the beams are expanded to a $e^{-2}$ diameter of $d \sim 1.7$ cm and are circularly polarized in the same sense by a pair of $\lambda/4$ wave plates. The timing sequence for the experiment is illustrated in Figure 3b. A mechanical shutter on the upper platform closes before the read-out pulse in order to block the back-scatter of stray read-out light produced by various optical elements. This light would otherwise interfere with the coherent signal from the atoms. A gated photo-multiplier tube (PMT, $8 \times 10^{-5}$ W/V at 780 nm, noise equivalent power 100 nW) is used to detect the power in the back-scattered field. Figure 3c shows an example of the echo signal recorded by the PMT averaged over 16 repetitions of the two-pulse AI. This signal is converted to units of optical power and numerically integrated to obtain a quantity which we term the echo energy. This quantity is proportional to the contrast of the atomic density modulation and the intensity of read-out light incident on the atoms. As a result, the signal is sensitive to both atom number fluctuations and photon number shot noise. This is a drawback compared to fluorescence detection techniques, where the optical transition can be saturated and is therefore less sensitive to photon shot noise [65]. In these experiments, we typically observe a noise floor of 0.1 pJ per shot, or $\sim$0.025 pJ after averaging over 16 repetitions, which was dominated primarily by the NEP of the PMT.

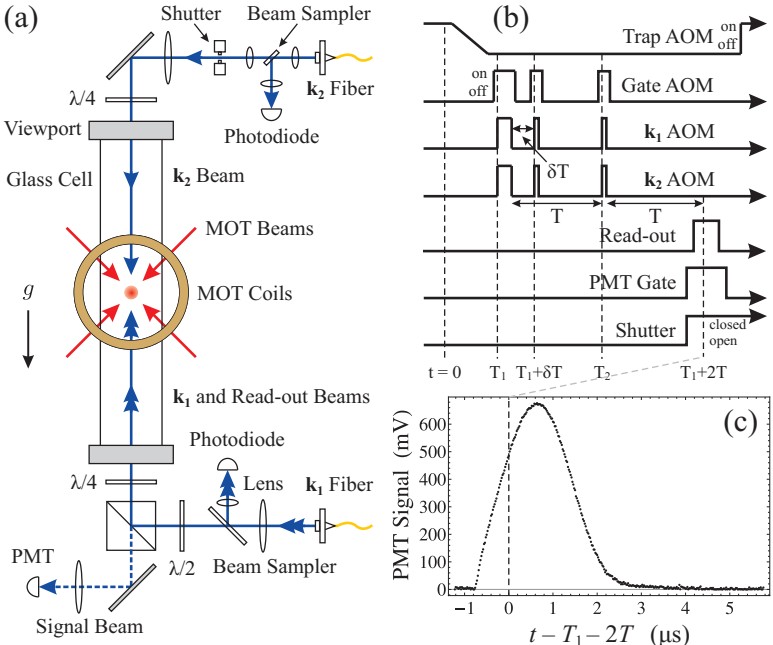

**Figure 3.** (**a**) Optical setup for the interferometer. The glass cell has dimensions $7.6 \times 7.6 \times 84$ cm and is oriented along the vertical. (**b**) Timing diagram for the AI. The gate acousto-optic modulator (AOM) is pulsed on to allow light for each excitation pulse produced by the $k_1$ and $k_2$ AOMs. The pulse occurring at $t = T_1 + \delta T$ corresponds to the third standing wave (SW) pulse. The read-out pulse (which is independent of the gate AOM) and the photo-multiplier tube (PMT) gate are turned on for $\sim$9 $\mu$s in the vicinity of the echo time, $t = T_1 + 2T$. (**c**) Example of a two-pulse grating-echo signal (from a 10 $\mu$K $^{87}$Rb sample) recorded by the PMT, which corresponds to an echo energy of 130 pJ. AI pulse spacing: $T = 1.06338$ ms; pulse durations: $\tau_1 = 3.8$ $\mu$s, $\tau_2 = 1.2$ $\mu$s; AI and read-out beam detunings: $\Delta_{AI} = 220$ MHz, $\Delta_{RO} = 40$ MHz; AI and read-out beam intensity: $I \sim 40$ mW/cm$^2$.

*2.4. Results*

Measurements of $\omega_q$ were obtained using the modified three-pulse AI by measuring the echo energy as a function of the third pulse time, $\delta T$, as illustrated in Figure 4a. This figure shows a measurement of $\omega_q$ in $^{85}$Rb on a time scale of $T \simeq 36.7$ ms, which was acquired in 15 minutes. Clearly, the shape of the fringes does not resemble that predicted by the theory shown in Figure 2(c). This is due to the contribution from each of the magnetic sub-levels in the $F = 3$ ground state of $^{85}$Rb, which tend to smear out the higher harmonics in the signal as a result of their different optical coupling strengths. Furthermore, the presence of additional, nearby excited states ($F' = 2$ and 3 in the case of $^{85}$Rb) produces an asymmetry in the fringe lineshape [56]. This effect is reduced in $^{87}$Rb because the frequency difference between neighboring excited states is larger. To measure $\omega_q$, the data are fit to a phenomenological model that consists of a periodic sum of exponentially-modified Gaussian functions

$$F(\delta T; \tau_q) = \sum_l A_l \exp\left[\frac{1}{2}\left(\frac{\sigma_l}{v}\right)^2 + \frac{\delta T - l\tau_q}{v}\right] \mathrm{erfc}\left[\frac{1}{\sqrt{2}}\left(\frac{\sigma_l}{v} + \frac{\delta T - l\tau_q}{\sigma_l}\right)\right], \qquad (18)$$

and the recoil frequency, $\omega_q = \pi/\tau_q$, is extracted from the fit. In this model, $\mathrm{erfc}(x)$ is the complementary error function, and the parameter $v$, which determines the amount of asymmetry in the lineshape, is the same for all fringes. The fit to the data shown in Figure 4a yielded a reduced chi-squared of $\chi^2/\mathrm{dof} = 0.51$ for dof $= 300$ degrees of freedom. This corresponds to a relative statistical precision of $\sim$180 ppb in $\omega_q$—representing a factor of $\sim$9 improvement over previous work [100].

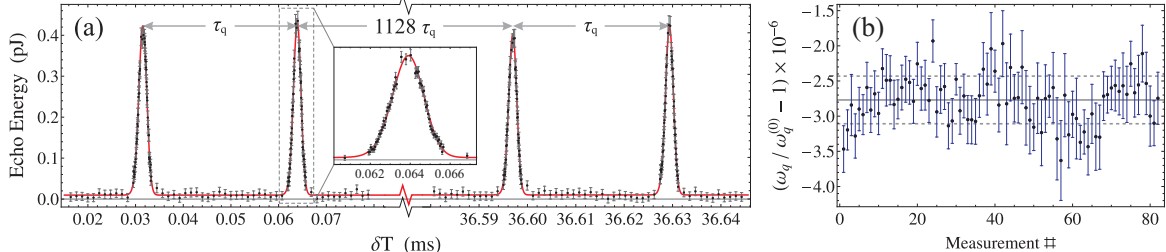

**Figure 4.** (**a**) Demonstration of an individual recoil measurement in $^{85}$Rb using the modified three-pulse AI. Data were recorded in two temporal windows separated by $T = 1132\,\tau_q \simeq 36.67$ ms. A least-squares fit to the data yields a statistical uncertainty in $\omega_q$ of 180 ppb. Inset: expanded view of the fringe near $\delta T = 64$ $\mu$s. (**b**) 82 chronological measurements of $\omega_q$ in $^{87}$Rb using $T \simeq 45.5$ ms. Each measurement was acquired in 10 minutes and produced a typical statistical error of $\sim$380 ppb. We have scaled these results by the expected value of the recoil frequency, $\omega_q^{(0)} = 94.77384783(12)$ rad/ms, which is based on the value of $h/M(^{87}\mathrm{Rb})$ from Ref. [15] and the $F = 2 \rightarrow F' = 3$ transition frequency in $^{87}$Rb from Ref. [103]. The dashed grid lines indicate the weighted standard deviation of 339 ppb, and the standard deviation of the mean is 37 ppb. The corresponding reduced chi-squared is $\chi^2/\mathrm{dof} = 0.93$ for dof $= 81$ degrees of freedom. The mean value, shown by the solid grid line, is $\sim$2.8 ppm below the expected value, which is due to systematic effects. AI pulse parameters: $T = 45.4837$ ms, $\tau_1 = 2.2$ $\mu$s, $\tau_2 = 1.4$ $\mu$s, $\tau_3 = 3$ $\mu$s, $\Delta_{\mathrm{AI}} = 219.8$ MHz, $\Delta_{\mathrm{RO}} \sim 40$ MHz, $I \sim 95$ mW/cm$^2$.

To demonstrate the long-term statistical sensitivity of the interferometer, 82 independent measurements of $\omega_q$ in $^{87}$Rb were recorded (see Figure 4b) while holding all experimental parameters fixed to the extent possible. Here, $\omega_q$ is determined from a weighted average over all individual measurements, where the points are weighted inversely proportional to the square of their statistical errors. The mean value shown in Figure 4b, which has not been corrected for systematic effects, is found with a statistical uncertainty of 37 ppb as determined by the standard deviation of the

mean. An autocorrelation analysis of these data indicate that they are correlated at the 20% level with measurements taken at a previous time. This is attributed to slowly-varying environmental conditions, such as temperature and magnetic field, over the 14 hours of data acquisition.

### 2.5. Discussion of Systematic Effects

We have investigated systematic effects on the measurement of $\omega_q$ related to the angle between excitation beams, the refractive indices of the sample and the background Rb vapor, light shifts, Zeeman shifts, *B*-field curvature and the SW pulse durations [56]. The total systematic uncertainty in this measurement is estimated to be $\sim$5.7 parts per million (ppm), and is dominated by two effects: (i) the refractive index of the sample, and (ii) the curvature of the *B*-field that the atoms experience as they fall under gravity. We now discuss these two effects in detail.

The refractive index of the atomic sample affects the wave vector of the excitation beams, since a photon in a dispersive medium acts as if it has momentum $n\hbar k$, where $n$ is the index of refraction [104]. For near-resonant light, the index becomes a function of both the density of the medium, $\rho$, and the detuning of the applied light from the atomic resonance, $\Delta_{AI}$. The systematic effect on the recoil frequency due to the refractive index can be expressed as $\omega_q(\rho, \Delta_{AI}) = \omega_q^{(0)} n^2(\rho, \Delta_{AI})$, where $\omega_q^{(0)}$ is the recoil frequency in the absence of systematic effects. The index of refraction can be computed from the electric susceptibility and the light-induced polarization of the medium [104]. Taking into account the level structure of the atom, it can be shown that [56]:

$$n(\rho, \Delta_{HG}) = \sqrt{1 - \frac{\rho}{\epsilon_0 \hbar \Gamma} \sum_H \mu_{HG}^2 \frac{\Delta_{HG}/\Gamma}{1 + (\Delta_{HG}/\Gamma)^2}}. \tag{19}$$

Here, $\Delta_{HG} \equiv \omega - (\omega_H - \omega_G)$ is the atom-field detuning between the ground and excited manifolds, $|g, G\rangle$ and $|e, H\rangle$, for laser frequency $\omega$, $G$ ($H$) is a quantum number representing the total angular momentum of a particular ground (excited) manifold, and $\mu_{HG}$ is the reduced dipole matrix element for transitions between those manifolds [74]. The root-mean-squared density of the cold sample at the time of trap release was estimated to be $(4.1 \pm 1.2) \times 10^{10}$ atom/cm$^3$ based on time-of-flight images [56]. Hence, we estimate a refractive-index-induced shift in $\omega_q$ of $-10.5 \pm 3.0$ ppm at a detuning of $\Delta_{AI} = 220$ MHz.

The other dominating systematic effect is due to the inhomogeneity of the magnetic field sampled by the atoms during the total interrogation time ($2T \sim 91$ ms) of the interferometer. This field primarily originates from nearby ferromagnetic material, such as an ion pump magnet and a glass-to-metal adaptor, and from the set of quadrupole coils we use to cancel the residual field in the vicinity of the MOT [56]. At the end of the optical molasses cooling phase, the atoms are distributed roughly equally in population among the magnetic sub-levels of the upper ground state $|F = 2\rangle$. A spatially-varying *B*-field with a non-zero curvature (i.e., $\beta_2 \equiv \partial^2 B/\partial z^2 \neq 0$) has a parasitic impact on the interferometer as a result of a position-dependent force on the $m_F \neq 0$ states similar to a harmonic oscillator (We have also considered the effect of a constant background *B*-field, and found that it produces a small systematic effect on $\omega_q$ ($\sim$7.5 ppb per Gauss of residual *B*-field) as a result of the distribution of sub-level populations. Similarly, linear magnetic gradients do not affect the measurement of $\omega_q$ using the echo AI [32].). For each momentum state trajectory, the atom samples a different region of space and experiences a different acceleration than that of a neighbouring trajectory. Since the momentum of each trajectory is differentially modified between excitation pulses, the interference for each class of trajectories occurs at a slightly different time—causing both a systematic shift of $\omega_q$ and a loss of interference contrast. For a given state $|F, m_F\rangle$, the corresponding systematic correction to $\omega_q$ is

$$\omega_q(\beta_2, T) = \omega_q^{(0)} \left[1 + \frac{2}{3} \left(\frac{m_F g_F \mu_B \beta_2}{M}\right) T^2\right]. \tag{20}$$

where $g_F$ is a Landé g-factor, and $\mu_B$ is the Bohr magneton. In the worst case scenario, the systematic shift in the recoil frequency is dominated by the state $|F = 2, m_F = 2\rangle$. Based on measurements of the $B$-field in the vicinity of the atoms, we estimate $\beta_2 \sim 0.1$ mG/cm$^2$ = $10^{-4}$ T/m$^2$. Thus, for $T = 50$ ms we estimate a relative shift of $\sim 11$ ppm. A more detailed analysis [56,57], which accounts of the distribution of magnetic sub-level populations, yields a more realistic estimate of $6.3 \pm 4.4$ ppm.

Including other minor systematic effects, such as the relative beam angle ($-15 \pm 8$ ppb), the refractive index of the background vapor ($-52 \pm 31$ ppb), light shifts due to the interferometer beams and the saturated absorption setup ($-55 \pm 2$ ppb), and the finite SW pulse durations ($0 \pm 2$ ppm), we estimate the total systematic shift on our measurement of $\omega_q$ to be $-4.3 \pm 5.7$ ppm [56]. Correcting for this shift, we find that our measurement $\omega_q = 94.77400(54)$ rad/ms is within 1.6 ppm of the expected value of $\omega_q^{(0)} = 94.77384783(51)$ rad/ms, as derived from the most precise measurement of $h/M$ [15]. The combined statistical (37 ppb) and systematic (5.7 ppm) uncertainties of our measurement are enough to account for this discrepancy.

We now discuss techniques for reducing the aforementioned systematic effects. Equation (19) for the refractive index suggests that the relative correction to $\omega_q$ can be reduced by decreasing the sample density, $\rho$, or by increasing the excitation beam detuning, $\Delta_{AI}$. However, the current configuration of the AI relies on a large number of atoms to achieve a sufficient signal-to-noise ratio. Thus, a decrease in the sample density leads to a reduction in the signal size. Furthermore, the sensitivity of the three-pulse AI relies on a relatively strong atom-field coupling in order to excite many orders of momentum states. An increase in the excitation beam detuning without a corresponding increase in the field intensity leads to a reduction in the sensitivity to $\omega_q$. The refractive index systematic could be reduced by a factor of $10^3$ by a 10-fold reduction in the rms density of the sample, accompanied by a 100-fold increase in the detuning. This would require an increase in the excitation field intensity by a factor of 100 (corresponding to $\sim 10$ W/cm$^2$) in order to retain the same sensitivity to $\omega_q$. A more promising way forward is to utilize the frequency-dependence of the refractive index, which exhibits "magic" frequencies where the systematic shift cancels, as shown in Figure 5. These frequency are located between two excited state manifolds, where the dispersive corrections to the refractive index due to each state have the same magnitude but opposite sign. Since these frequencies are independent of the density $\rho$, they are ideal for cancelling the refractive index shift due to both the cold-atom sample and the background vapor [56]. Using this feature of the refractive index, one can avoid both reducing the sample density and using high-intensity excitation beams.

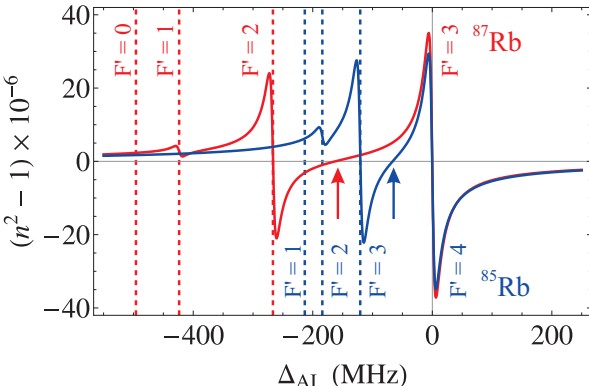

**Figure 5.** Relative correction to the recoil frequency due to the refractive index as a function of the detuning of the excitation field, $\Delta_{AI}$. These curves are based on Equation (19) with a density of $\rho = 10^{10}$ atoms/cm$^3$. Predictions for both $^{85}$Rb and $^{87}$Rb are shown. The detuning is plotted with respect to the $F = 3 \rightarrow F' = 4$ transition in $^{85}$Rb, and the $F = 2 \rightarrow F' = 3$ transition in $^{87}$Rb. The dashed grid lines label the location of excited states [103,105]. The "magic" frequencies, where the relative correction crosses zero, are indicated with arrows at $\Delta_{AI} \approx -66.4$ MHz for $^{85}$Rb and at $\Delta_{AI} \approx -162.6$ MHz for $^{87}$Rb.

The systematic shift due to the *B*-field curvature can be significantly reduced by preparing the sample in the magnetically "insensitive" state $|F = 2, m_F = 0\rangle$ (This can be achieved by employing a combination of optical pumping, resonant push beams and microwave $\pi$-pulses resonant with the $|F = 1, m_F = 0\rangle \rightarrow |F = 2, m_F = 0\rangle$ transition, which the standard technique in Raman interferometers and atomic clocks.). Then, any systematics due to the *B*-field would originate from the second-order Zeeman effect which shifts all sub-levels the $F = 2$ hyperfine manifold by a frequency $\frac{1}{2}KB^2$, where $K \simeq 575$ Hz/G$^2$ is the Zeeman shift of the clock transition in $^{87}$Rb [103]. Compared to the first-order Zeeman shift of $m_F g_F \mu_B \simeq 1.4$ MHz/G for the state $|F = 2, m_F = 2\rangle$, the corresponding force on the atom is reduced by many orders of magnitude for the same *B*-field strength. Utilizing only $m_F = 0$ atoms in the interferometer will have the added benefit of significantly reducing decoherence due to the *B*-field curvature—enabling an increase in *T* and a corresponding reduction in the statistical error of each measurement. Under current conditions, we have achieved a maximum time scale of $T \sim 65$ ms. However, previous studies indicate that the transit time of the atoms in the excitation beams is ∼270 ms [32], suggesting that *T* can be as large as ∼135 ms before the temperature of the sample becomes the limiting factor.

Another avenue for improvement to increase the signal-to-noise ratio in the experiment, which directly affects the sensitivity to $\omega_q$. As mentioned in Section 1.2, the macroscopic grating behaves as a linear reflector for an optical field of wavelength $\lambda$ with a complex reflectivity $r$ [85,86]. Using measurements of the energy in the back-scattered echo signal and the optical power in the read-out pulse, we estimate a reflection coefficient of $R = |r|^2 \sim 0.001$ under typical experimental conditions. The reflectivity can potentially be increased by pre-loading the sample in an optical lattice such that the initial spatial distribution has a significant $\lambda/2$-periodic component [106]. Experimental studies of MOTs loaded into an intense, off-resonant optical lattice have shown that the reflection coefficient of the light that is Bragg-scattered off the resulting atomic grating can be as large as $R \simeq 0.8$ [107]. This motivates the pursuit of high-contrast grating production using a far-detuned lattice pulse that precedes the AI excitations. Numerical simulations of the reflection coefficient from the grating produced by a lattice-loaded sample indicate that a 100-fold increase in the back-scattered signal is feasible. Such an endeavor would require an apparatus with good stability and control of the phase of the SW fields (i) to effectively channel atoms into the nodes of the lattice potential without significant heating, and (ii) to match the phases of the excitation and lattice fields.

By implementing these improvements to the echo AI, we anticipate that a future round of recoil measurements will yield results with both statistical and systematic uncertainties at competitive levels.

## 3. Measurements of Gravitational Acceleration

### 3.1. Overview of Gravity Measurements

Interest in precise measurements of the gravitational acceleration *g* have been stimulated in part by the connection of such measurements to the determination of the universal gravitational constant *G* [18–20,108] and the possibility of the variation of the gravitational force on small-length scales [109]. Since these measurements can be designed to measure the absolute value of *g* or relative changes due to temporal effects such as tides and positional variations due to changes in density, gravimeters have played a ubiquitous role in the exploration of natural resources by detecting characteristic density profiles associated with minerals, petroleum, and natural gas. An important practical consideration is the ability of these sensors to provide a non-invasive technique for exploration in wide area (air, sea, or submersible) mineral assays involving environmentally-sensitive areas. Other applications include borehole mapping for verifying properties of rocks, determination of bulk density for the detection of cavities, and tidal forecasts. The most precise relative measurements of *g* are derived from superconducting quantum interference-based devices (SQUIDs) [110,111], whereas absolute measurements of *g* based on an optical Mach-Zehnder interferometer [112] can achieve an absolute

accuracy of 1 ppb in an integration time of 20 minutes. This sensor relies on recording the chirped accumulation of fringes when a corner-cube retro-reflector on one arm of the interferometer falls through a height 0.3 m.

Interest in cold-atom-based interferometers began with the path breaking experiments in Refs. [8,113], which relied on a Raman interferometer to achieve a statistical precision of 3 ppm in a measurement time of 1000 s. Raman AIs have also obtained the most sensitive atom-based measurements of *g*. To select some examples, Ref. [114] achieved a statistical precision of 1.3 ppb in 75 s of data acquisition, whereas Ref. [22] included a detailed study of systematic effects and reported a statistical precision of 3 ppb over 1 min of integration. A key feature of both experiments was the active vibration stabilization of the inertial reference frame (i.e., the surface of the retro-reflection mirror) with respect to which the measurements were carried out. Additionally, in these examples atoms were launched in a 50 cm atomic fountain to obtain a free-fall time of over 300 ms. More recently, a Raman AI with a 6.5 m drop zone achieved an inferred single-shot sensitivity of $7 \times 10^{-12} \, g$ [36,37]. The Raman AI has also been developed to realize the best atom-based measurements of gravity gradients [23,24] and rotations [27–29,36,115]. As a consequence, this AI has been the preferred configuration for remote sensing applications [11,47,48,52,64,116–121].

Alternative techniques have also demonstrated competitive measurements of *g*. Experiments relying on Bloch oscillations report statistical precisions of 50–200 ppb after a few minutes of integration time [51,53], and 220 ppb of total systematic uncertainty after a few hours in the case of Ref. [122] (This latter measurement relies on the precision of Planck's constant *h*, which is presently known to 12 ppb [95].). Additionally, a single-state Mach-Zehnder interferometer involving Bragg transitions reported a sensitivity of 2.7 ppb after 1000 seconds of data acquisition using a drop height of 20 cm and passive vibration stabilization [52]. More recently, the same group demonstrated a Bragg-pulse gravimeter using a Bose-condensed sample which yielded an asymptotic uncertainty of 2.1 ppb [33]. The echo AI described in Ref. [58], which we review in this article, achieved a statistical precision of 75 ppb in one hour using a drop height of 1 cm and an apparatus in which only key components were passively vibration stabilized.

We now review measurements of *g* using echo AIs that rely on samples of laser-cooled rubidium atoms released from a MOT. The theoretical background and earlier results are described in Refs. [30,32,54,55,58,87,123].

### 3.2. Description of Echo AI Techniques for Measuring g

We first provide a discussion of the physical principles of AI configurations used for measurements of *g* that are based on the earlier theoretical description. Figure 6a represents the recoil diagram which shows displacements of centre-of-mass trajectories of wavepackets for momentum states for the two-pulse configuration of the AI based on a billiard ball model [124–126]. A sample of laser-cooled $^{85}$Rb or $^{87}$Rb atoms is excited along the vertical by two blue-detuned standing wave (SW) pulses separated by a time $T_{21}$. Each SW pulse is composed of two traveling wave components, each carrying a wave vector with wavenumber $k = 2\pi/\lambda$. Atoms in each of the magnetic sublevels of the $F = 3$ ground state in $^{85}$Rb or the $F = 2$ ground state in $^{87}$Rb are diffracted into a superposition of momentum states separated by $\hbar q$ at $t = 0$, where $q = 2k$. This process involves the absorption of a photon from one travelling wave component of the standing wave and stimulated emission along the counter-propagating traveling wave component. The durations of the SW pulses are sufficiently short that they meet the Raman-Nath criterion [127,128], where the displacement of the atoms due to the momentum transfer from standing wave pulses is small compared to the spacing of the quasi-sinusoidal standing wave potential. For counter-propagating traveling wave components, the wavelength of the potential is $\lambda/2$. The classical description of the effect of the standing wave interaction is that the atoms are focused toward the nodes of the standing wave potential. The focusing of atoms into the nodes produces a one-dimensional density grating with a period of $\lambda/2$. In the quantum mechanical description, it is the interference between momentum

states that produces this one-dimensional density grating. In this latter description, the atomic wave function develops a recoil modulation on a time scale $\tau_q = \pi/\omega_q \sim 32$ $\mu$s, where $\omega_q = \hbar q^2/2M$ is frequency associated with the two-photon recoil energy of the atom. The velocity distribution of the cold sample along the SW axis causes the grating to dephase on a much shorter time scale $\tau_{\text{coh}} = 1/ku$, where $u = \sqrt{2k_B\mathcal{T}/M}$ is the $1/e$ width of the velocity distribution and $\mathcal{T}$ is the temperature of the laser-cooled sample. For typical sample temperatures of $\sim 20$ $\mu$K, the dephasing time scale is $\sim 2$ $\mu$s. A long time $T_{21}$ after the grating has dephased, a second SW pulse is applied to diffract the momentum states. The effect of this SW pulse is to cause the momentum states separated by $\hbar q$ to rephase at the echo time $2T_{21}$. Momentum state interference produces a maximum contrast in the density grating just before and just after the echo time. The rephasing is reminiscent of a two-pulse photon echo experiment [129] that involves a superposition of ground and excited states. The echo technique is a general method of cancelling Doppler dephasing in an atomic gas. In echo atom interferometry, this technique has been extended to ground states so that velocity selection is not required. The effect of cooling the sample is simply to ensure that the time scale of the experiment is suitably long. Under ideal conditions, the experimental time scale is limited by the transit time of atoms across the excitation beam.

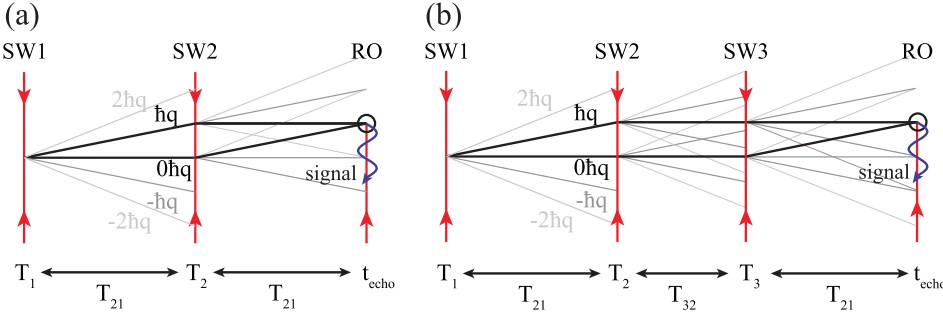

**Figure 6.** Recoil diagrams for (**a**) two-pulse and (**b**) three-pulse AIs in the absence of gravity. Only a subset of all trajectories are shown. SW refers to standing wave pulses and RO is a traveling wave read-out pulse. The SW pulses, composed of two counter-propagating traveling wave components, each with wave vector $|\mathbf{k}|$, diffract atoms into a superposition of momentum states separated by $\hbar q$. For both two-pulse and three-pulse AIs, the backscattered signal arises from interferences between states differing by $\hbar q$ at the echo time.

Since the atoms are in the ground state, it is necessary to apply a near-resonant, travelling-wave read-out pulse in the vicinity of the echo time to detect the contrast and phase of the re-phased density grating. The periodic array of atoms formed at the echo time coherently back-scatters the read-out pulse. This process is known as Bragg scattering. The grating spacing of $\lambda/2$ causes a total path difference change of $\lambda$ for light reflecting from adjacent planes of the grating. This effect produces constructive interference since the phase difference of reflections from adjacent planes is $2\pi$. In this case, the wavelength of the back-scattered light is matched with the Fourier component of the density grating with spacing $\lambda/2$. The back-scattered signal due to the read-out pulse is called the echo signal. To determine $g$, the phase of this coherent signal, which scales as $qgT_{21}^2$, is measured as a function of the pulse separation, $T_{21}$.

The read-out light is back-scattered as a consequence of the law of conservation of momentum. If an incoming read-out photon is backscattered, then the total momentum delivered to the sample is $\hbar q$. This momentum transfer allows the two arms of the interferometer that differ by $\hbar q$ to recombine. This action of the read-out pulse also creates a coherent superposition of ground and excited states throughout the sample. The total radiation pattern from this system of dipole radiators is phase-matched only along the backward direction. The experiment measures the phase of the echo signal with respect to an inertial frame of reference defined by an optical local oscillator (LO) with a

frequency $\omega_{LO}$. A convenient reference frame is defined by the nodal point of a standing wave, such as a reflecting surface.

The back-scattered light at frequency $\omega_{AI}$ is detected as a beat note at frequency $|\omega_{LO} - \omega_{AI}|$ using a heterodyne technique [55]. Although the quadratic accumulation of phase as a function of $T_{21}$ is appealing for a precision measurement of $g$ (among various interferometer geometries, this configuration also encloses the largest space-time area), the signal amplitude exhibits recoil modulation as well as a chirped frequency, resulting in the need for a complicated fit function to extract $g$. The signal from the two pulse AI is analogous to the interference fringes recorded by the falling corner-cube optical interferometer discussed in Ref. [112].

An alternate configuration involves a three-pulse stimulated echo AI [32,75,123,130], as shown in Figure 6b. Here, the first SW pulse creates a superposition of momentum states separated by $\hbar q$. A second SW pulse applied at $t = T_{21}$ produces momentum states that are co-propagating at a fixed spatial separation with the same momentum during a central time window of duration $T_{32}$. A third SW pulse applied at $t = T_{21} + T_{32}$ causes the co-propagating states to interfere at the echo time $t = 2T_{21} + T_{32}$, resulting in a density grating. Just like the two-pulse AI, the grating formation is associated with interference of momentum states separated by $\hbar q$. The signal amplitude as a function of pulse separation $T_{32}$ shows no recoil modulation and exhibits a fixed angular frequency $qgT_{21}$ due to the velocity $gT_{21}$ acquired by the atoms during the time interval $T_{21}$ in the presence of gravity. The period of the signal amplitude is given by $\tau_v = \lambda/2gT_{21}$.

The constant modulation period improves the quality of the fits to the data, thereby resulting in improved statistical precision in measurements of $g$. For our experimental conditions, in which the setup was partially shielded from vibrations, the stimulated echo AI proved particularly useful. This is because the time window $T_{21}$ can be made small compared to the time scale over which vibrations cause mirror positions to become uncorrelated, while $T_{32}$ can be relatively large, thereby realizing a larger total time scale than the two-pulse AI. This feature is due to the co-propagating momentum states that have a constant spatial separation during the time window $T_{32}$. The disadvantages include the reduction of the signal amplitude due to the additional SW pulse and the inherent sensitivity to any initial velocity along the SW axis.

### 3.3. Experimental Setup

Figure 7a shows a block diagram of the experimental setup. The details are described in [58]. A Ti:Sapphire laser is used to generate light for atom trapping and interferometry using a chain of acousto-optic modulators (AOMs) that serve as frequency shifters and amplitude modulators. All these elements are placed on a pneumatically-supported optical table. Light from these AOMs is transported to the atom trap using angle-cleaved, anti-reflection (AR) coated optical fibers.

The experimental setup for atom trapping and atom interferometry is shown in Figure 7b. The vacuum chamber used for atom trapping is made of 316 L stainless steel and it is anchored to an optical table mounted on pneumatic vibration isolators. The chamber is maintained at $5 \times 10^{-9}$ Torr by an ion pump with a pumping speed of 270 L/s located 1 m away to reduce ambient magnetic fields. The chamber is surrounded by three pairs of magnetic field and gradient cancelling coils. A separate set of tapered coils wound on the chamber provides the magnetic gradient for atom trapping. The trapping optics, vacuum chamber, anti-Helmholtz and cancellation coils, and ion pump are supported by the optical table. The MOT is loaded from background vapor, with approximately $5 \times 10^8$ atoms loaded in 1 second. Time-of-flight charge-coupled device (CCD) camera images of atoms released after molasses cooling [131] show that the typical sample temperature is 20 $\mu$K.

The fiber-coupled beam used for atom interferometry identified in Figure 7a is aligned through a single-pass AOM operating at 250 MHz, as shown in Figure 7b. The circularly-polarized diffracted beam from this AOM, which is directed along the vertical and used for excitation of atoms, is detuned by $\Delta \sim 55$ MHz above resonance [76]. This beam is retro-reflected through the atom cloud

by a corner-cube reflector to produce standing wave excitation. The undiffracted beam, with a frequency of $\omega_0 + 305$ MHz is spatially separated from the excitation beam by 2.5 cm. It is aligned through the same optical elements as the excitation beam to minimize the impact of relative phase changes due to vibrations and serves as an LO. The LO is physically displaced upon reflection by the corner-cube. The background light entering the apparatus during the AI pulse sequence is minimized by pulsing the gate AOM in Figure 7a only when the AI AOM is turned on. The excitation and LO beams, combined on a beam splitter and a balanced heterodyne detector with two oppositely-biased Si photodiodes with rise-times of 1 ns, are used to record a beat signal at a frequency $\omega_{\mathrm{RF}} = |\omega_{\mathrm{AI}} - \omega_{\mathrm{LO}}| = 250$ MHz. During the read-out pulse, the retro-reflection of the excitation beam is blocked by a mechanical shutter with an open/close time of 1 ms [132].

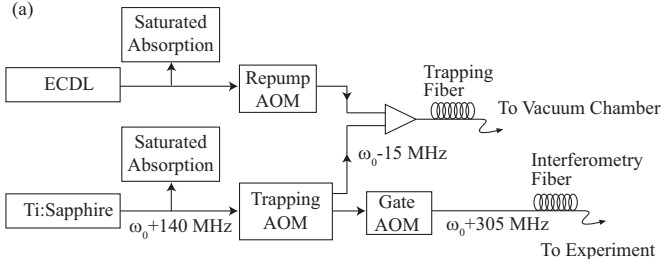

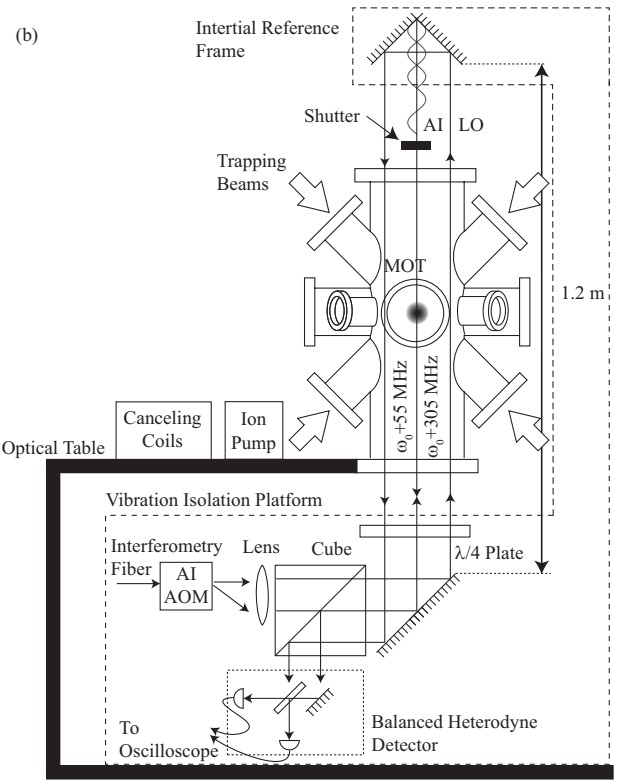

**Figure 7.** Schematic of experimental setup. (**a**) Block diagram of laser sources and frequency control. The frequency of the laser light is defined with respect to $\omega_0$, the resonant frequency of the ${}^{87}$Rb $F = 2 \rightarrow F' = 3$ transition or the ${}^{85}$Rb $F = 3 \rightarrow F' = 4$ transition. (**b**) Schematic of the atom interferometry setup. The lower-detection optics and the upper corner-cube reflector are anchored together and placed on a vibration isolation platform. The vacuum chamber and vibration isolation platform rest on an optical table supported by pneumatic legs. The photodiodes detect a 250 MHz beat note, which is the frequency difference between the AI and LO beams. The forms of the ion pump, cancelling coils, and anti-Helmholtz trapping coils are not shown.

The corner-cube reflector, AI AOM, balanced detector, and related optics are anchored to a vibration isolation platform with a resonance frequency of 1 Hz, which rests on the pneumatically-supported optical table, as shown in Figure 7b. The optical table is effective in suppressing vibration frequencies above 100 Hz, whereas the vibration isolation platform suppresses frequencies in the range of 1–100 Hz. The mechanical shutter is separately anchored to the ceiling of the laboratory to reduce vibrational coupling. In this setup, only critical components are passively isolated with the vibration isolation platform.

Digital delay generators with time bases controlled by a 10 MHz signal from a rubidium clock (Allan variance of $10^{-12}$ in 100 seconds) are used to produce RF pulses with an on/off contrast of 90 dB to drive the AOMs. The time delays of optical pulses are controlled with a precision of 50 ps. The read-out pulse intensity is comparable to the saturation intensity of Rb atoms so that the entire echo signal envelope can be recorded without appreciably decohering the signal. This signal, which is measured as a 250 MHz beat note, is recorded on an oscilloscope with an analog bandwidth of 3.5 GHz and mixed down to DC using the RF oscillator driving the AI AOM to produce the in-phase ($E_0 \cos(\phi_g)$), and in-quadrature ($E_0 \sin(\phi_g)$) components of the back-scattered electric field. While the atom trap is loaded, an attenuated excitation beam is turned on to record a 250 MHz beat note. This measurement re-initializes the RF phase used to mix the signal down to DC at the beginning of each repetition of the experiment and ensures that the relative phases between the excitation beam and the LO are the same at the start of the experiment. Although the LO and AI beams are strongly correlated at the beginning of the experiment, the phase uncertainty progressively increases with the time scale of the experiment and it cannot be corrected mainly because the motion of the corner-cube reflector is not measured. The typical repetition rate of the experiment varies between 0.8–3 Hz.

### 3.4. Theory

We now review the theoretical description of the signal shapes and characteristics for both two- and three-pulse stimulated AI configurations using simplified equations that apply to an atomic system with a single magnetic ground state sublevel as in Refs. [32,58,87]. Here, $g$ represents a constant gravitational acceleration along the axis of SW excitation.

### 3.4.1. Two-Pulse AI

The backscattered electric field due to the readout pulse for the two-pulse AI can be written as

$$E_g^{(2)} = E_0^{(2)} e^{i\phi_g^{(2)}}, \tag{21}$$

where $E_0^{(2)}$ is the electric field amplitude and $\phi_g^{(2)}$ is the gravitational phase. The electric field amplitude for the two-pulse AI can be shown to be

$$E_0^{(2)} \propto E_{RO} e^{-(\Delta t/\tau_{coh})^2} J_1 \left[ 2\theta_1 \sin(\omega_q \Delta t) \right] J_2 \left[ 2\theta_2 \sin\left(\omega_q (T_{21} + \Delta t)\right) \right] e^{-t_{echo}/\tau_{decay}}, \tag{22}$$

where $E_{RO}$ is the electric field amplitude of the read-out pulse, $J_n(x)$ is the $n^{th}$-order Bessel function of the first kind, $\theta_1$ and $\theta_2$ are the pulse areas of the first and second SW pulses respectively, $\Delta t = t - 2T_{21}$ is the time relative to the echo time $t_{echo} = 2T_{21}$, and $\omega_q = \hbar q^2/(2M)$ is the two-photon recoil frequency. Here, $\tau_{coh} = 1/ku$ is the coherence time due to Doppler dephasing that defines the temporal width of the signal shown in Figure 8a, where $u = \sqrt{2k_B \mathcal{T}/M}$ is the $1/e$ width of the one-dimensional velocity distribution along the excitation beams and $\mathcal{T}$ is the sample temperature. The last term in Equation (22) represents a phenomenological decay, with a time constant $\tau_{decay}$ that models signal loss due to decoherence mechanisms in the experiment (e.g., from spontaneous emission and the spatial curvature of the ambient magnetic field), as well as the transit time of cold atoms through the interaction zone defined by the excitation beams.

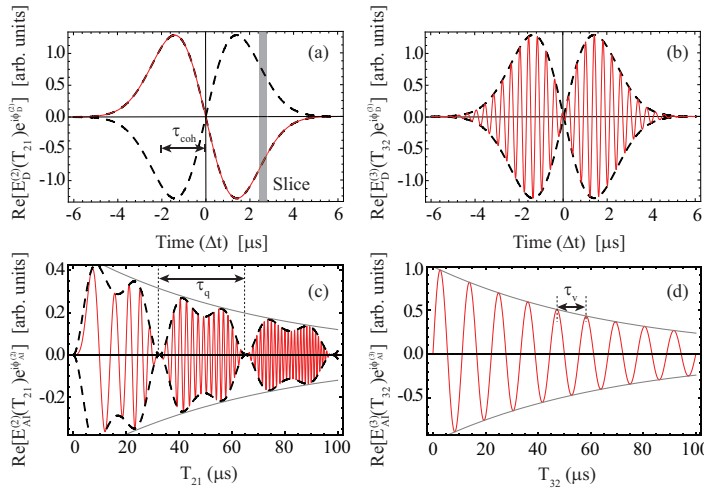

**Figure 8.** Predicted shape of the in-phase component of the echo envelope at (**a**) $2T_{21} = 0.1$ ms for the two-pulse AI and (**b**) $2T_{21} + T_{32} = 100$ ms for the three-pulse AI are shown as solid red lines. Here, $g = 9.8$ m/s$^2$, and $\Delta t$ is the time measured with respect to the echo time. In (a), a time slice used in the analysis is shown as a gray rectangle. The two echo envelopes shown in red are described by $E_{\mathrm{D}}^{(2)} e^{i\phi_{\mathrm{D}}^{(2)}}$ for the two-pulse AI (see Equations (25) and (27)), and $E_{\mathrm{D}}^{(3)} e^{i\phi_{\mathrm{D}}^{(3)}}$ for the three-pulse AI (see Equations (34) and (36)). The black dashed lines show the envelopes in the absence of gravity. The echo envelope exhibits an increase in the oscillation frequency as the free-fall time increases. (**c**) Predicted shape of the in-phase component of the signal amplitude for the two-pulse AI as a function of $T_{21}$ as predicted by $E_{\mathrm{AI}}^{(2)} e^{i\phi_{\mathrm{AI}}^{(2)}}$ (see Equations (26) and (28)), and shown as a solid red line. The signal exhibits chirped sinusoidal behaviour due to the quadratic dependence of $\phi_{\mathrm{AI}}^{(2)}$ on $T_{21}$, and also shows recoil modulation with a period $\tau_q = 33.151 \mu$s. Here, we set $g = 980$ m/s$^2$ for illustrative purposes. The black dashed lines show the total signal amplitude. (d) Predicted shape of the in-phase component of the signal amplitude for the three-pulse AI as a function of $T_{32}$ as predicted by $E_{\mathrm{AI}}^{(3)} e^{i\phi_{\mathrm{AI}}^{(3)}}$ (see Equations (35) and (37)), shown as a solid red line. The signal exhibits a constant modulation frequency with a period $\tau_v$ and shows no recoil modulation. Here, $T_{21} = 15$ ms. Again, we set $g = 980$ m/s$^2$ for illustrative purposes. The total signal amplitudes for both two-pulse and three-pulse AIs show a phenomenological exponential decay (gray line) to illustrate signal loss due to decoherence and transit time effects.

In the presence of gravity, the space-time area enclosed by the interferometer determines the phase accumulation and for the two-pulse AI [30,32,54,58,87], the phase is given by

$$\phi_g^{(2)} = \boldsymbol{q} \cdot \boldsymbol{g}(T_{21}^2 + 2T_{21}\Delta t + \Delta t^2/2). \tag{23}$$

We decouple the expression for the echo signal into two parts that are dependent on the time scales $T_{21}$ and $\Delta t$ to explain its characteristics. Accordingly, the two-pulse signal becomes

$$E_g^{(2)} = E_{\mathrm{D}}^{(2)}(\Delta t) E_{\mathrm{AI}}^{(2)}(T_{21}) e^{i\phi_{\mathrm{D}}^{(2)}(\Delta t)} e^{i\phi_{\mathrm{AI}}^{(2)}(T_{21})}, \tag{24}$$

where

$$\phi_{\mathrm{D}}^{(2)}(\Delta t) = \boldsymbol{q} \cdot \boldsymbol{g}(2T_{21}\Delta t + \Delta t^2/2) \tag{25}$$

is the Doppler phase which is dependent on $\Delta t$, and

$$\phi_{\mathrm{AI}}^{(2)}(T_{21}) = \boldsymbol{q} \cdot \boldsymbol{g} T_{21}^2 \tag{26}$$

is the AI phase which is dependent only on $T_{21}$. The Doppler component of the electric field amplitude is given by

$$E_{\text{D}}^{(2)}(\Delta t) \propto E_{\text{RO}} e^{-(\Delta t/\tau_{\text{coh}})^2} J_1 \left[ 2\theta_1 \sin(\omega_q \Delta t) \right], \tag{27}$$

and in the limit $\Delta t \ll T_{21}$, the AI electric field amplitude is given by

$$E_{\text{AI}}^{(2)}(T_{21}) = J_2 \left[ 2\theta_2 \sin(\omega_q T_{21}) \right] e^{-t_{\text{echo}}/\tau_{\text{decay}}}. \tag{28}$$

The measurement of gravity is based on detecting the amplitude and phase of the back-scattered light from the atomic grating (which has a frequency $\omega_{\text{AI}}$ and phase $\phi_{\text{AI}}$) with reference to an optical LO, which has a fixed frequency $\omega_{\text{LO}}$ and phase $\phi_{\text{LO}}$. The signal is recorded as a beat note at the frequency $\omega_{\text{AI}} - \omega_{\text{LO}}$ and with a phase difference $\phi_{\text{signal}} = \phi_{\text{AI}} - \phi_{\text{LO}}$. The phase shifts associated with the atoms are sensitive to optical phase shifts of the SW pulses and the LO due to the environment. The total signal amplitude can be expressed in terms of the in-phase and in-quadrature components $E_0^{(2)} \cos(\phi_g^{(2)})$ and $E_0^{(2)} \sin(\phi_g^{(2)})$ as

$$E_0^{(2)} = \frac{1}{\sqrt{2}} \left\{ \left[ E_0^{(2)} \cos(\phi_g^{(2)}) \right]^2 + \left[ E_0^{(2)} \sin(\phi_g^{(2)}) \right]^2 \right\}^{1/2}. \tag{29}$$

The recoil modulation and signal decay terms can be removed from the in-phase and in-quadrature components of the back-scattered field amplitude by normalizing with respect to $E_0^{(2)}$. We are then left with $\cos(\phi_g^{(2)})$ and $\sin(\phi_g^{(2)})$ as the two components of the signal.

The dashed lines in Figure 8a show the Doppler electric field amplitude as predicted by Equation (27). Here, a convenient $T_{21}$ was chosen to maximize the recoil modulated signal, modeled by Equation (28). The solid red line shows gravity-induced oscillations within the echo envelope, as predicted by $E_{\text{D}}^{(2)} e^{i\phi_{\text{D}}^{(2)}}$. The oscillations are attributed to the free fall of atoms through a grating spacing of $\lambda/2$, which results in a phase increment of $2\pi$. This effect can also be described as a Doppler shift of the backscattered field due to the falling grating.

The solid red line in Figure 8c shows the predicted in-phase component of the signal amplitude for the two-pulse AI as a function of $T_{21}$ given by $E_{\text{AI}}^{(2)} e^{i\phi_{\text{AI}}^{(2)}}$ (see Equations (26) and (28)). The recoil modulation its readily apparent and the frequency-chirped oscillations due to gravity are illustrated by setting $g = 980 \text{ m/s}^2$. The dashed black line shows the recoil-modulated total signal amplitude $E_{\text{AI}}^{(2)}$.

### 3.4.2. Three-Pulse AI

Based on Refs. [32,58,87], the backscattered electric field for the three-pulse stimulated echo AI shown in Figure 6b can be written as

$$E_g^{(3)} = E_0^{(3)} e^{i\phi_g^{(3)}}, \tag{30}$$

where $E_0^{(3)}$ is the electric field amplitude and $\phi_g^{(3)}$ is the gravitational phase. The electric field amplitude can be shown to be:

$$\begin{aligned} E_0^{(3)} &\propto E_{\text{RO}} e^{-(\Delta t/\tau_{\text{coh}})^2} J_1 \left[ 2\theta_1 \sin(\omega_q \Delta t) \right] \\ &\times J_1 \left[ 2\theta_2 \sin\left(\omega_q(T_{21} + \Delta t)\right) \right] J_1 \left[ 2\theta_3 \sin\left(\omega_q(T_{21} + \Delta t)\right) \right] e^{-t_{\text{echo}}/\tau_{\text{decay}}}. \end{aligned} \tag{31}$$

Here, $\theta_3$ is the pulse area of the third SW pulse and the time relative to the echo time is $\Delta t = t - 2T_{21} - T_{32}$. This signal exhibits recoil modulation as a function of $T_{21}$ but not as a function of $T_{32}$.

In the presence of gravity, the phase of the three-pulse stimulated echo signal can be shown to be

$$\phi_g^{(3)} = \boldsymbol{q} \cdot \boldsymbol{g}(T_{21}^2 + T_{21}T_{32} + T_{32}\Delta t + 2T_{21}\Delta t + \Delta t^2/2). \tag{32}$$

As in the two-pulse case, this phase is proportional to the space-time area enclosed by the interferometer. Setting $T_{32} = 0$ reduces $\phi_g^{(3)}$ to the earlier result for $\phi_g^{(2)}$.

To explain the characteristics of the echo signal, we once again decouple the prediction into a part that is dependent on the time scales $T_{21}$, $T_{32}$, and a second part that is dependent on $\Delta t$. Therefore, the three-pulse echo signal is written as

$$E_g^{(3)} = E_D^{(3)}(\Delta t)E_{AI}^{(3)}(T_{21}, T_{32})e^{i\phi_D^{(3)}(\Delta t)}e^{i\phi_{AI}^{(3)}(T_{21}, T_{32})}, \tag{33}$$

where

$$\phi_D^{(3)}(\Delta t) = \boldsymbol{q} \cdot \boldsymbol{g} \left[ (T_{32} + 2T_{21}) \Delta t + \Delta t^2/2 \right] \tag{34}$$

is the Doppler phase, and

$$\phi_{AI}^{(3)}(T_{21}, T_{32}) = \boldsymbol{q} \cdot \boldsymbol{g}(T_{21}^2 + T_{21}T_{32}) \tag{35}$$

is the AI phase. The Doppler component of the electric field amplitude is given by

$$E_D^{(3)}(\Delta t) \propto E_{RO}e^{-(\Delta t/\tau_{coh})^2} J_1 \left[ 2\theta_1 \sin(\omega_q \Delta t) \right], \tag{36}$$

and the AI electric field amplitude is given by

$$E_{AI}^{(3)}(T_{21}, T_{32}) = J_1 \left[ 2\theta_2 \sin\left( \omega_q(T_{21} + \Delta t) \right) \right] J_1 \left[ 2\theta_3 \sin\left( \omega_q(T_{21} + \Delta t) \right) \right] e^{-t_{echo}/\tau_{decay}}. \tag{37}$$

We note that $\phi_D$ can be varied by changing either $T_{32}$ or $T_{21}$. As in the two pulse AI, this term produces a modulation of the echo envelope due to gravitational acceleration. Since $t_{echo} = 2T_{21}$ for the two-pulse AI and $t_{echo} = 2T_{21} + T_{32}$ for the three-pulse AI, the functional forms of $\phi_D^{(2)}$ and $\phi_D^{(3)}$ are in fact identical.

The functional forms of the electric field amplitudes in Equation (22) and Equation (31) are similar. However, since the three-pulse stimulated AI amplitude involves the additional experimental parameter $T_{32}$, the envelope of this echo signal can be recorded as a function of $T_{32}$ for an optimized value of $T_{21}$. For non-zero $T_{32}$, $\phi_{AI}^{(3)}$ is maximized if $T_{32} = 2T_{21}$.

The dashed lines in Figure 8b show the Doppler electric field amplitude predicted by Equation (36). This shape is plotted by choosing a convenient $T_{21}$ to maximize the recoil-modulated signal, modelled by Equation (37). The solid red line shows oscillations within the echo envelope due to gravity, as predicted by $E_D^{(3)}e^{i\phi_D^{(3)}}$.

The solid red line in Figure 8d shows the shape of the in-phase component of the signal amplitude for the three-pulse stimulated echo AI as a function of $T_{32}$, as predicted by $E_{AI}^{(3)}e^{i\phi_{AI}^{(3)}}$ (see Equations (35) and (37)). This signal exhibits a characteristic period $\tau_v$ determined by $T_{21}$ and shows no recoil modulation. The total signal amplitude $E^{(3)}(T_{21}, T_{32})$ predicted by Equation (37) as a function of $T_{32}$ is shown in Figure 8d as a grey line. This curve exhibits a smooth decay due to signal loss arising from transit time and decoherence effects.

## *3.5. Results*

### 3.5.1. Doppler Phase Measurements

The characteristics of the echo envelope can be used to measure $g$ along the axis of SW excitation. Equation (25) for the two-pulse AI and Equation (34) for the three-pulse stimulated echo AI show that the Doppler phase $\phi_D$ produces a similar modulation of the echo envelope for the two configurations.

The echo envelope has a simple dispersion shape shown in Figure 8a, and predicted by Equation (27) if $T_{21}$ and $T_{32}$ are small. As $T_{21}$ and $T_{32}$ are increased, the signal envelope exhibits oscillations due to gravity as shown by the single sequence acquisitions in Figure 9a,b for the two-pulse and three-pulse stimulated echo configurations, respectively. The oscillations due to $g$ are evident for the echo time $2T_{21} = 9.3$ ms in Figure 9a. In Figure 9b, the echo time $2T_{21} + T_{32} = 45.1$ ms with $T_{21} = 1.5$ ms. The increase in modulation frequency within the echo envelope as a function of $T_{21}$ for the two-pulse AI (see Equations (25) and (27)) and as function of $T_{32}$ for the three-pulse AI (see Equations (34) and (36)) were used in Refs. [58,87] to measure $g$ with a precision ranging from 0.6% to 0.8%. For these measurements, the Doppler modulation frequency across the echo envelope was assumed to be a constant since $T_{21}, T_{32} \gg \Delta t$ and the frequency was determined from eight repetitions. Although the relatively short measurement time scale is appealing, the sensitivity to the fit functions used to model the echo envelope and the temporal duration of the signal (a few microseconds) limited the statistical uncertainty. The utility of this technique can be re-examined in an actively vibration-stabilized apparatus that is discussed later in this section.

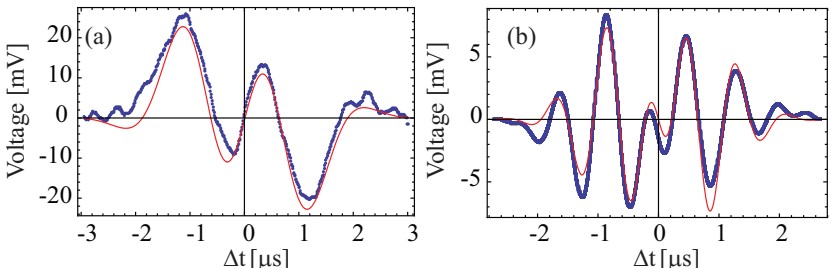

**Figure 9.** (**a**) Example of fit to the in-phase component of the two-pulse echo signal obtained on a single acquisition for $2T_{21} = 9.3$ ms. (**b**) Example of fit to the in-phase component of the three-pulse echo signal obtained on a single acquisition for $2T_{21} + T_{32} = 45.1$ ms.

### 3.5.2. Two-Pulse AI Measurement

The best method of obtaining amplitudes of the in-phase and in-quadrature components of the signal requires fitting to the signal shape as in Figure 9, but the consistency of the results is affected by the complicated fit functions. As a result, the signal from the two pulse AI is usually background subtracted, and the points are squared and summed over the signal duration to extract the two components of the signal amplitude as a function of either $T_{21}$. The quadrature sum of the component amplitudes gives the total signal amplitude $E_0^{(2)}$, and each of the component amplitudes is normalized with respect to $E_0^{(2)}$ to obtain $\cos(\phi_{AI}^{(2)})$ and $\sin(\phi_{AI}^{(2)})$. Although this procedure is suitable for extracting $\phi_{AI}^{(2)}$ predicted by Equation (26), it is particularly sensitive to background subtraction and the signal strength, and ignores the frequency variation across the echo envelope predicted by Equation (23). For each value of $T_{21}$, the signal amplitude $E_0^{(2)}$ is calculated by averaging over three successive points on either side. This procedure ensures that the sinusoidal fits are not skewed by the scatter in the signal strength.

Figure 10 shows a measurement of $g$ using the quadratic dependence of $\phi_{AI}^{(2)}$ on $T_{21}$ as predicted by Equation (26). The amplitude of the in-phase component is recorded as a function of $T_{21}$ for one hour with four observational windows, with each window consisting of 200–325 points acquired in a randomized sequence. Each data point represents the average of 16 repetitions. The error bars are determined on the basis of a probability density function (PDF) analysis [120]. The weights of the error bars are assigned as the product of the error bars from the PDF analysis and the error bars based on the signal amplitude $E_0^{(2)}$. The overall time scale was limited to $T_{21} = 12.8$ ms due to the progressive breakdown of the periodically initialized RF phase. These data show the expected chirped frequency dependence of $\cos(\phi_{AI}^{(2)})$ on $T_{21}$. The data are fit to a multi-parameter function of

the form $\cos(qgT_{21}^2 + qv_0T_{21} + \phi_0)$, which yielded a measurement of $g = 9.79123(9)$ m/s$^2$. Similarly, we obtain $g = 9.79130(9)$ m/s$^2$ from the in-quadrature component with a similar uncertainty. A weighted average of the in-phase and in-quadrature components allowed the acceleration to be determined with a statistical precision of 7 ppm. In the fit function, $v_0$ models a velocity parameter for the atoms, and $\phi_0$ is the initial phase of the grating with respect to the nodal point on the corner-cube reflector. The typical value of $v_0$ from the fit was 0.107(1) mm/s, which was much smaller than results obtained by tracking the centroid of the falling cloud with a CCD camera. Since cloud launch does not affect the phase of the two-pulse AI, we speculate that intensity imbalances in the two SW components produce this effect.

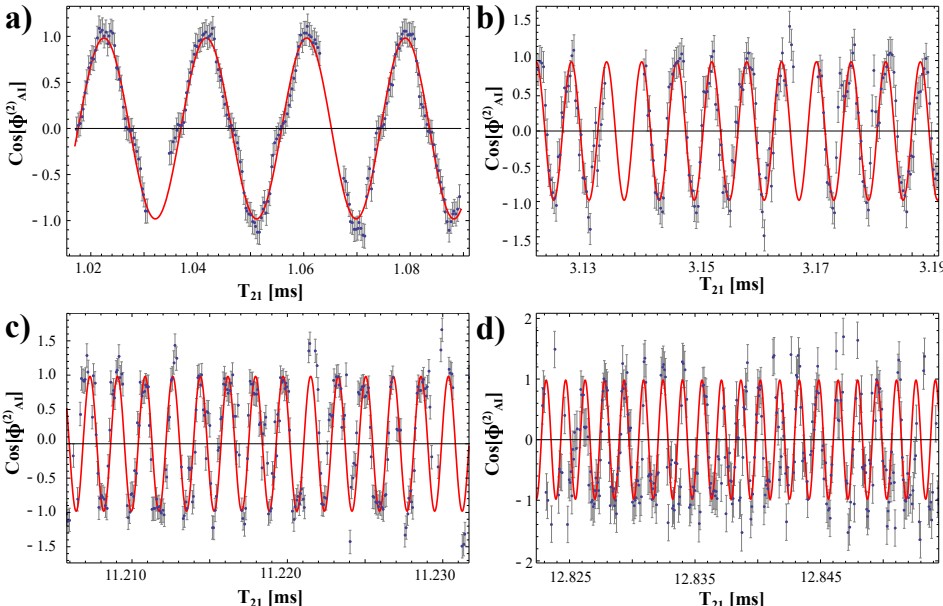

**Figure 10.** (**a–d**) Four observational windows for the two-pulse AI experiment showing the amplitude of the in-phase component as a function of $T_{21}$. The frequency of the signal increases linearly as a function of $T_{21}$ ($\partial\phi_{AI}^{(2)}/\partial T_{21} = 2qgT_{21}$). The solid red line indicates a simultaneous least-squares fit to all four windows. These results were obtained using $^{85}$Rb, with SW pulse lengths $\tau_1 = 800$ ns and $\tau_2 = 90$ ns, optical intensity of 50 mW/cm$^2$, at a detuning of $\Delta = 55$ MHz relative to the $F = 3 \rightarrow 4$ cycling transition.

We note that the size of the residuals in all four windows of Figure 10 is smaller than the standard deviation for a random distribution of points [58]. Here, the standard deviation of the histogram of the residuals in phase units for the entire data sets is 0.7 rad out of a total phase accumulation of $|\phi_{AI}^{(2)}| \sim 2.6 \times 10^4$ rad for $T_{21} = 12.85$ ms. The increasing size of the residuals for $T_{21} > 10$ ms in Ref. [58] indicates the sensitivity of the two-pulse AI to vibrations and decoherence effects such as magnetic field curvature.

### 3.5.3. Three-Pulse Measurement

We now discuss the improvements obtained using the three-pulse stimulated echo AI. Due to the relative insensitivity of the three-pulse AI to vibrations compared to the two-pulse AI, two previously-mentioned analysis techniques that have distinct disadvantages, namely fitting to the echo envelope, as well as the faster square-sum method can be avoided. Instead, the instantaneous amplitude of the background-subtracted signal is found from a single time slice of the echo envelope, as shown in Figure 8a. The best statistical precision was obtained with a temporal slice duration of 10 ns in which there is effectively no change in the signal amplitude. The average amplitude of

each slice was determined by averaging 16 repetitions. This method ensures that the signal amplitude can be measured for 200 slices over the the echo envelope for each value of $T_{32}$.

Figure 11a,b show $\cos(\phi_g^{(3)})$ for a single slice as a function of $T_{32}$ with $T_{21}$ fixed at 7.431900 ms. The in-phase and in-quadrature components $\cos(\phi_g^{(3)})$ and $\sin(\phi_g^{(3)})$ were obtained by normalizing with respect to the total signal amplitude $E_0^{(2)}$ as for the two-pulse AI. These data were recorded in one hour with 100 points in each window acquired in randomized sequence. The data shows that the signal exhibits a single frequency as predicted by Equation (35). Two widely spaced observational windows allow this frequency to be determined precisely. The frequency for a single slice can be written as

$$\omega_{\text{AI}}^{(3)} = \left| \frac{\partial \phi_{\text{AI}}^{(3)}}{\partial T_{32}} \right| = \boldsymbol{q} \cdot \boldsymbol{g} T_{21}. \tag{38}$$

Here, the frequency change across the echo envelope predicted by Equation (32) is ignored. We use Equation (38) to fit data for the in phase component with $T_{21} = 7.431900$ ms and $q = 16105651.65$ rad/m [103], to obtain $g = 9.833245(4)$ m/s$^2$, which represents a statistical precision of 0.4 ppm. This value can be compared to the 7 ppm statistical uncertainty for the two-pulse AI.

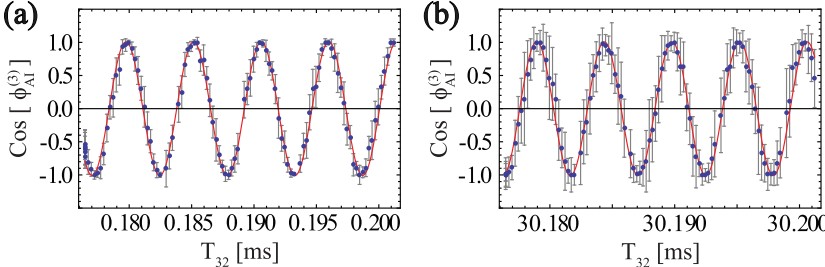

**Figure 11.** Two observational windows of the amplitude of the in-phase component for the three-pulse AI with $T_{21} = 7.4319$ ms as a function of $T_{32}$ in the vicinity of (**a**) $T_{32} = 0.2$ ms and (**b**) $T_{32} = 30$ ms. These data correspond to a single time slice and exhibit a constant frequency as a function of $T_{32}$ ($\partial \phi_{\text{AI}}^{(3)}/\partial T_{32} = qgT_{21}$). The solid red line indicates a simultaneous least-squares fit to both data windows, which yields a frequency of 187324.75(8) Hz. An analysis of the fit residuals indicates the standard deviation of phase noise for the entire data set is 0.2 rad. These results were obtained using $^{87}$Rb.

The enhancement in precision can be attributed to several factors. Firstly, the determination of a single frequency in the absence of recoil modulation increases the robustness of fits. Secondly, the measurement time scale is significantly extended in comparison to the two-pulse AI since these data represent a total time scale $2T_{21} + T_{32} = 45$ ms, while limiting $T_{21}$ to ~7.5 ms. Therefore, there is reduced sensitivity to the effects of magnetic curvature and vibrations, which also leads to a more gradual decay of the signal amplitude. As a result, the standard deviation of the residuals (~0.11) is similar in each observational window. In phase units, the standard deviation of the residuals for the entire data set is 0.2 rad. In comparison, the overall standard deviation is 0.7 rad for the two-pulse AI. The results show that the slicing technique works well only because of good phase stability.

The slicing technique also allows the frequency across the echo envelope predicted Equation (32) to be observed. Additionally, a further improvement in statistical uncertainty is achieved by determining the frequencies of all time slices across the echo envelope. Figure 12a shows the frequency of each time slice obtained using two widely spaced observational windows as in Figure 11 as a function of $\Delta t$. Here, the echo envelope is divided into 200 slices, each with a duration of 10 ns. The data confirm the linear dependence of the angular frequency on $\Delta t$ predicted by

$$\omega_a^{(3)} = \left| \frac{\partial \phi_a^{(3)}}{\partial T_{32}} \right| = \boldsymbol{q} \cdot \boldsymbol{g}(T_{21} + \Delta t), \tag{39}$$

with $\phi_a^{(3)}$ given by Equation (32). Each data point in Figure 12a has a typical error bar of 1 ppm. The reduction in error in comparison to the two-pulse AI is a result of the extended (30 ms) time scale. The improved sensitivity makes it possible to observe the change in frequency with $\Delta t$ across the echo envelope. In contrast, for the data associated with measurements of *g* using the Doppler phase modulation inside the echo envelope, we assume a constant frequency across the echo envelope since the observational window is only a few microseconds long and the measurement does not have the desired sensitivity.

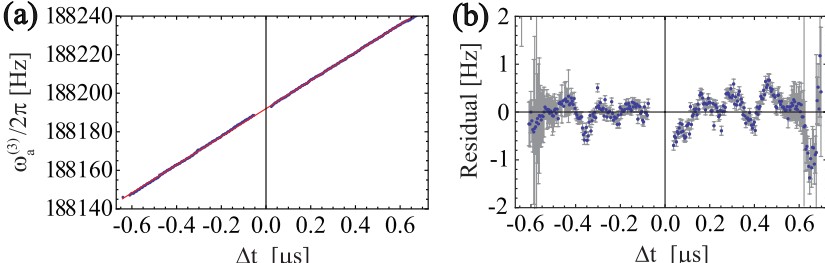

**Figure 12.** (**a**) Measurement of $\omega_a^{(3)}$ across the echo envelope using 200 time slices. Here, we use $T_{21} = 7.431900$ ms. The intercept of the line is 188192.095(17) Hz, giving a value of $g = 9.8774458(9)$ m/s$^2$ and a corresponding statistical error of 90 ppb (not corrected for systematic effects). Similar analysis of the in-quadrature component reduces the combined statistical error to 75 ppb. (**b**) Residuals of the linear fit. These results were obtained using $^{87}$Rb.

For the data in Figure 12a, the limited time scale of the echo envelope and the scatter lead to an overall statistical error in the slope that is appreciable (600 ppm) despite the relatively small statistical error in each of the points (1 ppm). The scatter is attributed to magnetic field effects described in Refs. [58,87]. However, the uncertainty in the frequency intercept is much more tightly constrained since the data are closely clustered near $\Delta t = 0$. Based on Equation (39), the frequency intercept is $\boldsymbol{q} \cdot \boldsymbol{g} T_{21}$. From the linear fit in Figure 12a, we find the intercept at $\Delta t = 0$ to be 188192.095(17) Hz. Using the values of *q* and $T_{21}$, we obtain $g = 9.8774458(9)$ m/s$^2$, which represents a statistical precision of 90 ppb. A weighted average from the in-phase and in-quadrature components gives a combined statistical precision of 75 ppb. Figure 12b shows the residuals to the straight line fit in Figure 12a. The residuals increase in size in the regions where the echo envelope is small such as in the extremities and in the vicinity of $\Delta t = 0$.

### 3.6. Improvements and Future Work

To summarize the gravity measurements described here, we note that an analysis of the Doppler phase oscillations of the echo envelope resulted in measurements statistically precise to 0.6%. Experiments with the amplitude and phase of the two-pulse AI yielded a statistical uncertainty of 7 ppm. Experiments with the three-pulse AI with a drop height of 1.2 cm in a passively vibration-stabilized chamber demonstrated the best statistical precision of 75 ppb, but it is currently not competitive with the precision of interferometers based on two-photon Raman or Bragg transitions [22,36,46–48,52,114].

The time scale of the experiments described here was principally limited by the magnetized vacuum chamber. The magnetization of the chamber and a correction due to the index of refraction were the dominant sources of systematic errors [58]. Other constraints were associated with the signal-to-noise ratio due to the power available from the laser source. The index correction, which impacts *q*, is dependent on both the sample density and the detuning of the excitation [104], and it is prominent because of the near-resonant nature of the experiment. Based on the improved signal-to-noise ratio obtained in recent echo experiments that utilized a non-magnetic apparatus [32,57], we have projected a measurement sensitivity of 0.6 ppb for the three-pulse AI and 0.3 ppb for the

two-pulse AI. Such an experiment can be realized with drop heights of 30 cm in a non-magnetic apparatus with active stabilization of the inertial reference frame. Such an experiment will also require excitation beams with a power output of several Watts so that far-off resonant excitation is feasible. This seemingly challenging requirement can be addressed by using low-cost laser systems consisting of tapered amplifier waveguides seeded by auto-locked diode laser systems [61] that are discussed in Section 5. It is expected that the new laser source will produce a ten-fold increase in signal size due to excitation of higher-order momentum states. Active vibration stabilization will allow cold atoms to be preloaded into a one-dimensional optical lattice so that the initial spatial distribution of atoms has a significant $\lambda/2$-periodic component [106]. This approach will produce gratings with reflectivities approaching unity [107], which would represent an appreciable increase in the signal-to-noise ratio compared to current experiments in which the reflectivity is $\sim 0.001$. This increase will permit the experiment to be carried out at a lower density to further reduce the refractive index correction. Atoms will also be selected in the magnetically "insensitive" $m_F = 0$ state to avoid systematic effects due to magnetic interactions. A reduction in measurement time will be realized by using techniques for under-sampling fringe patterns [133] developed for commercial corner-cube gravimeters. We now discuss the layout of the improved apparatus.

The main limitation in the passively-stabilized experiment discussed here is the lack of reference to a proper inertial frame. Since the interferometer's SW excitation beam is generated by retro-reflecting a traveling-wave beam off of a corner-cube retro-reflector, the phase of the SW excitation was linked to the position of the corner-cube with respect to Earth's gravity field. This means that during the course of the experiment, there would be an uncontrolled phase accumulation in the signal proportional to the drift of the position of the apparatus. This puts an upper limit on the possible time scale of experiments, which in turn limits the achievable precision. Depending on the vibration spectrum of the lab environment, there might also be aliasing effects for certain experimental repetition rates, as suggested by the data in Refs. [58,87].

The new apparatus shown in Figure 13 will include a high-precision accelerometer, attached to the corner-cube reflector to provide a more reliable reference of the experiment's frame of reference to the Earth's frame. The position of the reflector as a whole can be measured at the beginning of each experiment, and this data can be used to either post-correct the data, or to actively correct the frame of reference using a mechanical actuator such as a voice-coil [22,24]. The sensitivity of the interferometer to motion of the SW potential during the experiment can be easily calculated [40,134], making the post-correction process relatively straightforward.

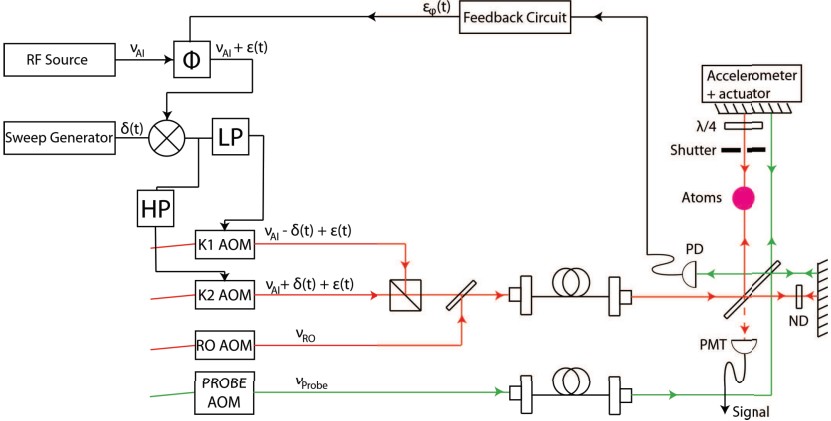

**Figure 13.** Improved apparatus to measure of $g$ using an echo AI.

The apparatus will be passively-isolated from ambient vibrations by a two-stage system. The first stage consists of a laminar flow pneumatic isolator, on which the optics table is mounted. This isolator has a vertical resonance around 1 Hz. To further decrease the system's susceptibility

to low-frequency vibrations, the atom trapping optics, AI optics, and vacuum chamber are mounted on a passively-tuned spring damping system that has a sub-1 Hz vertical resonance. This isolator is, in turn, mounted on top of the previously-mentioned pneumatic isolation system. The only component of the apparatus that is not mounted to the spring-based isolator is the field cancellation coils, due to the mass limit of the spring-based isolator. The coils are instead attached directly to the pneumatically-isolated table. Since the zero field/gradient region created by the cancelling coils is quite large compared to the expected motion of the apparatus due to vibrations, the mounting scheme for the coils is not expected to affect the stability of the measurements.

Because the AI relies on two traveling wave components to produce SW excitation, it is necessary to monitor the drift of the retro-reflecting mirror with respect to the other optics in the system using a Michelson-type interferometer and a far-off resonance probe beam. One arm of this interferometer will include the retro-reflecting mirror and the polarizing-cube beam splitter along the path of the vertical excitation beam path and the output of the interferometer will be recorded on a sensitive photodetector. This signal from the probe will be used to postcorrect the AI phase measurement. Such a correction will account for the movement of the lower optics with respect to the retro-mirror. Since the retro-mirror's position with respect to Earth will be monitored by a sensitive accelerometer, the phase of the SW excitation with respect to Earth will be fully determined. In an alternative scheme, the signal from the interferometer could be used to actively control the position of the retro-mirror with respect to the lower optics through the aforementioned voice coil and a PID control loop [8,22].

Another important shortcoming of the apparatus used in this work related to the magnetic field canceling coils. In the absence of state selection, the presence of magnetic field gradients and magnetic field curvature in the vicinity of the atoms during the experiment results in an additional force, which cannot be separated from the gravitational force. As such, the available time scale for experiments was also limited by the incomplete magnetic field cancellation and the magnetization of the vacuum chamber. The cancellation was incomplete due to the non-optimal geometry of the field-canceling coils and the magnetization of the vacuum chamber. In brief, the B-field at the position of the atoms is canceled by 3 sets of mutually-perpendicular coils. Optimal cancellation is obtained when each set of coils is arranged in a "Helmholtz" configuration, where the (circular) coils are separated by a distance equal to their radius. In our experiment, the coils were set up in a pseudo-Helmholtz configuration, with square coils separated by their side length. This resulted in a well-controlled field only in the immediate vicinity of the atom trap, with no extended volume of zero field. The new apparatus will make use of three sets of square cross-section Helmholtz coils, whose ideal distance of separation was found using numerical modeling to be 0.55 times the side length. These coils produce an extended volume of zero magnetic field, allowing the atoms to experience free-fall for several hundred milliseconds. If the atoms are launched in a fountain, the experimental time scale can be further extended to 1 s. Extension of time scales beyond 50 ms used in this work is also expected to require chirped standing wave excitation to avoid loss of signal amplitude due to the differential Doppler shift of falling atoms with respect to the traveling wave components of the standing wave. This limitation can be overcome using chirped, counter-propagating traveling-wave beams to cancel the Doppler shift due to falling atoms [32,57].

## 4. Magnetic Coherences

Properties of atomic coherences have been exploited for interesting applications such as quantum state preparation [135] and control [136], nondemolition measurements [137,138], entanglement [139], and precision magnetometry [140,141]. Other interesting applications have related to studies of velocity-changing collisions using both radio frequency (RF) [142] and optical excitation [143] and experiments with coherent transient effects that involve coupling between Zeeman sublevels [144,145]. In this section, we review applications of coherent transient techniques developed for atom interferometry for precise measurements of the strength of magnetic interactions such as atomic g-factor ratios [59,60].

For the experiments considered in Ref. [60], the sample of atoms is excited using two simultaneous traveling-wave laser pulses applied at $t = 0$ with wave vectors $\vec{k}_1$ and $\vec{k}_2$ at a small angle $\theta \sim 10$ mrad, as shown in Figure 14a. The atomic sample is a cloud of laser-cooled atoms loaded into a MOT. The individual traveling waves pulses have orthogonal linear or circular polarizations and are detuned from the excited state. However, the pulses are resonant with the two-photon transition that couples two magnetic sublevels of the ground state, as shown in Figure 14b. The timing diagram is shown in Figure 14c. The excitation creates a spatially-periodic superposition between the magnetic sublevels of the ground state coupled by the laser fields. The superposition has a period of $\lambda/\theta$, where $\lambda$ is the wavelength of the excitation. This coherence grating is probed by a read-out pulse along $\vec{k}_2$ and the resulting signal, referred to as magnetic-grating free induction decay (MGFID) [146], is coherently scattered along $\vec{k}_1$ due to conservation of momentum. The grating dephases due to the thermal motion of the atoms, and the time scale of the signal decay is determined by the time taken by a typical atom to move a distance on the order of a grating spacing $(\lambda/\theta u)$, where $u$ is the most probable speed associated with the Maxwell-Boltzmann velocity distribution. A magnetic grating echo (MGE) is observed using a second set of excitation pulses at $t = T$ to rephase the coherence grating, as in Figure 14c [146–148]. The second pulse modifies the time-dependent coefficients that describe the coherent superposition of magnetic sublevels so that the grating reforms at $t = 2T$. This is analogous to the reversal of the Doppler phases of individual atoms in a traditional two-pulse photon echo experiment [149] that can be used for measuring the excited state lifetime [150]. In the absence of decoherence due to collisions and background light, the MGE amplitude should decay on a time scale determined by the transit time of atoms through the laser beams [148].

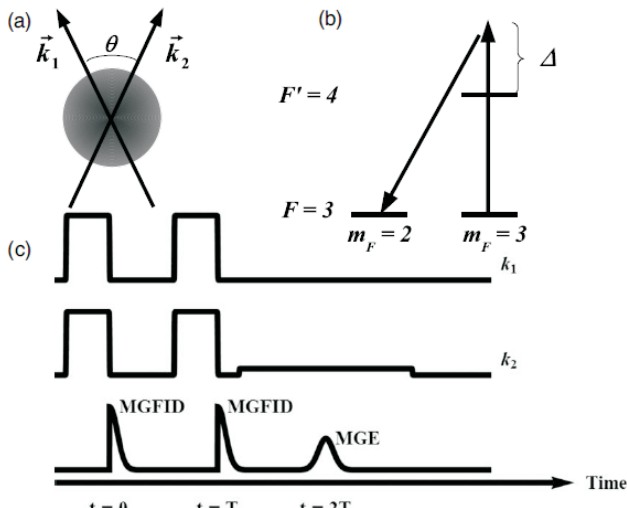

**Figure 14.** (**a**) Laser pulses along $\vec{k}_1$ and $\vec{k}_2$ excite a sample of laser-cooled atoms; $\theta \sim 10$ mrad. (**b**) Level diagram for the experiment with $\vec{k}_1$ and $\vec{k}_2$ having orthogonal linear polarizations; detuning $\Delta \sim 40$ MHz. (**c**) Timing diagram for the magnetic grating echo (MGE) signal.

The focus of initial experimental work using the MGE was related to observing effects due to atomic recoil [147]. The MGFID and MGE were used to verify the expected dependence of the dephasing time of the coherences on the velocity distribution of the sample [147,148], observing the effect of magnetic fields for particular experimental configurations [147], and studying the effects of collisions [148]. Other applications of the MGFID include measurements of the diffusion constants [151] and phase space imaging [152]. Echo techniques were also used to investigate applications related to detecting nanostructures and depositing periodic arrays of atoms on substrates [153].

In a static magnetic field, the Zeeman shift between magnetic sublevels causes temporal oscillations within the envelopes of the MGFID and MGE signals at multiples of the Larmor frequency $\omega_L$. In Ref. [60], an analytical calculation was developed to predict the functional form of the Larmor oscillations in the MGFID for arbitrarily-directed weak static magnetic fields in situations in which the excitation pulses consisted of traveling waves with orthogonal linear and circular polarizations. This theoretical treatment is based on a rotation matrix approach [154–156], in which the effect of the magnetic field can be described as a time-dependent rotation of the atomic system about the quantization axis. This treatment accurately described signals observed from room temperature vapor and laser-cooled atoms. In the absence of magnetic fields, the velocity distribution of a cold sample measured using the MGFID agreed with the results of an independent technique used to measure the sample temperature [131]. Additionally, a rate equation treatment [157,158] was used to understand the properties of the MGE in a magnetic field. Thus, Ref. [60] renewed interest in precise measurements of atomic g-factor ratios using the MGFID and MGE signals and compact laser-cooled samples that could be excited in uniform magnetic fields.

The best measurements of Zeeman shifts [159] and atomic g-factor ratios [160] were carried out with uniform $\sim$50 G *B* fields in centimeter-length, paraffin-coated atomic vapor cells to avoid decoherence due to wall collisions. These experiments relied on RF spectroscopy and lamp-based optical pumping to determine the centers of transitions between atomic ground states of Rb isotopes. The typical accuracies of a few ppm were sufficient for independent tests of the non-relativistic Zeeman Hamiltonian. However, a measurement of g-factor ratios precise to 500 ppb or better can be used to test corrections arising from the self-energy of the electron and vacuum polarization. These effects have been incorporated into atomic theory describing the electron and nuclear g-factors $g_J$ and $g_I$, respectively [161], but have not been verified by experiments. The predicted corrections to $g_J$ and $g_I$ depend on the nuclear mass and spin, which of course differ among isotopes. Measurements of isotopic atomic g-factor ratios therefore constitute a sensitive test of specific aspects of QED that relate to magnetic interactions. The importance of such experiments for matter-antimatter comparisons has been highlighted by the recent measurements of the magnetic moment of the antiproton [162].

Although measurements of atomic g-factor ratios were undertaken using the MGFID [54], the statistical precision was overwhelmed by systematic effects due to AC Stark shifts. Therefore, an alternative technique, which is the basis of atomic magnetometers [163] was adopted in Ref. [59] to demonstrate the most sensitive measurement of g-factor ratios. Our statistical precision of 690 ppb, obtained on a 10-ms time scale, exceeded the 2 ppm sensitivity of a long-standing measurement [160]. In our experiment, Larmor oscillations of ground state coherences from a dual isotope MOT containing $^{85}$Rb and $^{87}$Rb atoms were simultaneously recorded. The measurement exploited the compact dimensions of the sample (few millimeters) over which a uniform, actively-stabilized magnetic field was applied. Coherences involving $F = 3$ ($F = 2$) hyperfine ground states in $^{85}$Rb ($^{87}$Rb) were optically excited with a circularly-polarized pulse with a wave vector perpendicular to a constant, weak ($B < 1$ G) magnetic field. Coherences between adjacent magnetic sublevels in each isotope can be modeled as a magnetization that precesses in the *B* field at the Larmor frequency $\omega_L = g_F \mu_B B / \hbar$. The precession can be measured with a linearly-polarized probe pulse directed orthogonal to the *B* field. A rotation in the polarization of the probe pulse due to the differential absorption of its $\sigma^+$ and $\sigma^-$ components can be recorded on a balanced detector. This signal exhibits sinusoidal oscillations at $\omega_L$. Therefore, simultaneous measurements of $\omega_L^{85}$ and $\omega_L^{87}$ as a beat note (see Figure 15) can be used to measure the ratio of Lande $g_F$-factors.

The precision of 690 ppb and data acquisition time (3 minutes) allowed systematic effects due to Breit-Rabi corrections and AC Stark shifts to be investigated [59]. Results showed that the nonlinear variation of $\omega_L$ on *B* due to the Breit-Rabi effect did not cancel when the ratio of $g_F$-factors was measured, resulting in a systematic shift that depended on *B*. The shift also depended on the magnetic quantum number $m_F$. Therefore, parameters of the probe pulse and *B* fields affected the

weighted sum (of pairs) of coherences across the ground state manifolds that contribute to the signal. These systematic effects were also qualitatively confirmed using numerical simulations.

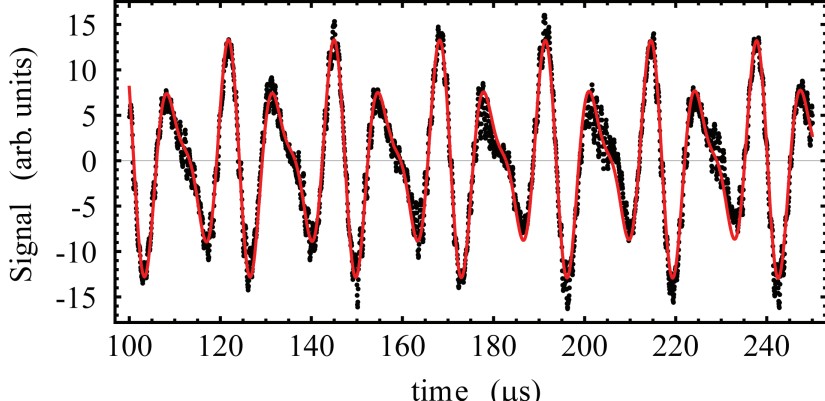

**Figure 15.** Ground-state coherence signal from dual-isotope magneto-optical trap (MOT) showing multiple-frequency components in the signal. Only one eighth of the points are displayed so that the fit (red line) and the data (black dots) can be distinguished. From a single shot, the ratio of effective atomic g-factors $r = g_F{}^{87}/g_F{}^{85}$ is typically determined to a precision of ∼1.5 ppm.

Subsequently, atom interferometric experiments that shared the same setup [32,57] achieved even better control of *B*-fields and *B*-gradients resulting in observation times of ∼250 ms—limited only by the transit time of cold atoms. Such an experiment with a 100-ms time scale will be ideally suited for this measurement. The experiment will resolve Larmor oscillations associated with all pairs of coherences in the ground state manifolds of the two isotopes by using a larger, uniform, pulsed *B* field of about 5 G. Additionally, Larmor oscillations from coherences in the $F = 2(F = 1)$ ground states of $^{85}$Rb ($^{87}$Rb) will be measured in addition to signals from the $F = 3(F = 2)$ ground states in $^{85}$Rb ($^{87}$Rb). This approach will eliminate Breit-Rabi corrections and allow the measurement of $g_J$ and $g_I$. The estimated measurement precision of ∼100 ppb will permit tests of predicted relativistic corrections at the desired level [161]. The possibility of realizing similar measurements with RF excitation and optical pumping techniques can also be investigated.

## 5. Auto-Locked Lasers

We have presented a review of recent results pertaining to measurements of $g$, atomic recoil frequency, and atomic g-factor ratios. Although all these experiments have recorded significant improvements in precision, the measurements are dominated by systematic effects such as the index of refraction due to the near-resonant excitation using laser beams with power outputs of ∼100 mW. We envision control of systematic effects at the ppb level by using far-detuned excitation beams with relatively high-power outputs of a few Watts. Ideally, several high-power lasers will be required for atom trapping, atom interferometry, and lattice loading—a crucial step for increasing the contrast (and signal-to-noise ratio) of density gratings in future echo interferometer experiments. Commercial laser sources such as Ti:Sapphire lasers with output powers of ∼1 W are inadequate apart from being expensive to maintain over the data collection times of several thousand hours. A possible option is to use frequency-doubled light from erbium-doped fiber-amplified laser sources operating in the 1560 nm telecommunications band [164,165], with output powers of >10 W. However, the cost is comparable to Ti:Sapphire lasers and the lifetime of these sources has not been tested beyond a few years.

To fulfill experimental requirements, we have relied on elegant and practical designs for external cavity diode lasers (ECDLs) and controllers [166–171] that produce power outputs of 1-100 mW and linewidths of ∼1 MHz, to develop unique, low-cost, vacuum-sealed, auto-locked external cavity

diode laser systems (ALDLS) that can be integrated using components from original equipment manufacturers (OEM), specially machined parts, and powerful central processors [61]. The laser source depends on optical feedback from a narrow-band interference filter to realize a narrow laser linewidth and improved spatial mode quality. The thermally-stabilized laser cavity can be evacuated within minutes and vacuum-sealed for several months, so that laser system exhibits reduced sensitivity to environmental temperature and pressure fluctuations. These lasers can be locked or scanned with respect to atomic or molecular spectral lines without the need for human intervention using a digital controller. The laser cavity relies on an interchangeable optics kit consisting of a laser diode and optical feedback elements to operate in the desired wavelength range. The digital signal processor of the controller is capable of storing a variety of algorithms in its memory for laser frequency stabilization using techniques such as pattern matching and first or third derivative feedback. Additional features include power amplification to several Watts using semiconductor waveguides [172], frequency locking to external cavities, and rapid amplitude modulation for wide-ranging applications. These capabilities are not commercially available as a package despite the widespread availability of the constituent OEM components.

In recent trials, ALDLS systems, operating at both 780 nm and 633 nm, have proven their linewidth ($\sim$300 kHz) and lock stability ($\sim$500 kHz) through accurate measurements of gravitational acceleration $g$ using a state-of-the-art industrial gravimeter (Scintrex FG5X) with an absolute accuracy of 1 ppb. The value of $g$ determined using our prototype lasers was in agreement with the baseline established using an iodine-stabilized He-Ne laser (which is an industrial standard), and exhibited lower scatter. These results suggest that systematic effects in cold-atom measurements of $g$ [58] can be characterized using an industrial gravimeter and a common ALDLS-based light source operating at 780 nm. Three improvements that were targeted have also been realized in recent tests. Firstly, the vacuum-sealed cavity reduced frequency drifts by a factor of fifty, as shown in Figure 16, so that the lock stability could be extended to 24-hour time scales. Figure 17 shows the Allan deviation of a typical OEM current controller used to operate the ALDLS. Our results suggest that significant reduction in current noise is possible, since the best current noise density specification achieved in commercial controllers (which use a Libbrecht-Hall design [171]) is $\sim$200 pA/$\sqrt{\text{Hz}}$. Secondly, interchangeable optics kits used in the same laser head have allowed operation at two widely-separated wavelengths accessible to laser diodes. Thirdly, the controller demonstrated the use of different algorithms for locking to atomic and molecular resonances. In the 780 nm band, the control system relied on atomic rubidium spectroscopy. To lock the laser, the controller implemented pattern matching between Doppler-free spectra obtained in real time by scanning the laser and the reference peaks stored in the processor's memory by using a sliding correlation algorithm. This technique determines the frequency offset between the scanned pattern and the reference pattern. The offset is fed back as a control voltage to the piezo transducer that controls the length of the laser cavity. As a result, the laser frequency is iteratively brought closer to the desired frequency with variable scan amplitude. In the 633 nm band, the controller achieved tighter locks by generating the third derivatives of narrower molecular resonances in iodine. Additionally, the controller was automatically able to relock even in the event of a mode hop, because it could be programmed to vary the diode current and temperature and compare spectral peaks over a scan range of up to 10 GHz. The availability of tapered amplifier waveguides with power outputs of several Watts, and techniques to transiently increase the drive current [173] to achieve a several-fold enhancement in output power, suggest that suitable low-cost laser systems can be developed for pursuing echo experiments in a far-off resonance configuration.

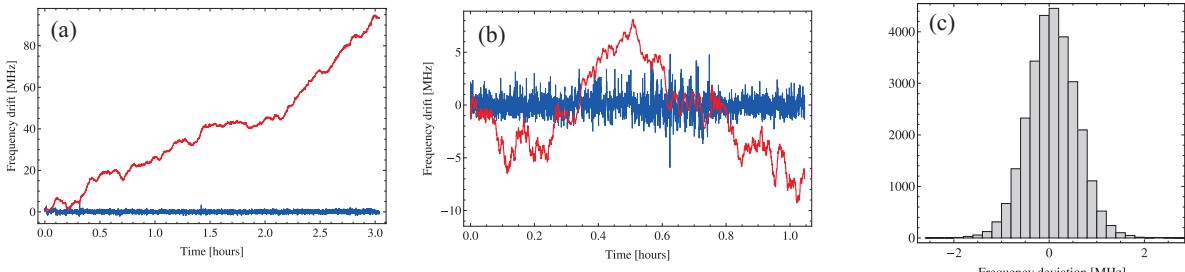

**Figure 16.** (**a**) Correction signal (red) and lock signal (blue) obtained with laser auto-locked to an iodine transition. The laser head was maintained at atmospheric pressure so that pressure-induced drifts of the correction signal extend over nearly 100 MHz. (**b**) Same signals as in part (a) with the laser head pumped down to 1 mTorr. The correction signal is uncorrelated with pressure changes and the range of frequency drifts shows a tenfold reduction. (**c**) Typical histogram of lock signals (blue curves in parts (a) and (b)) exhibits a full width at half maximum of 500 kHz, which is a measure of lock stability.

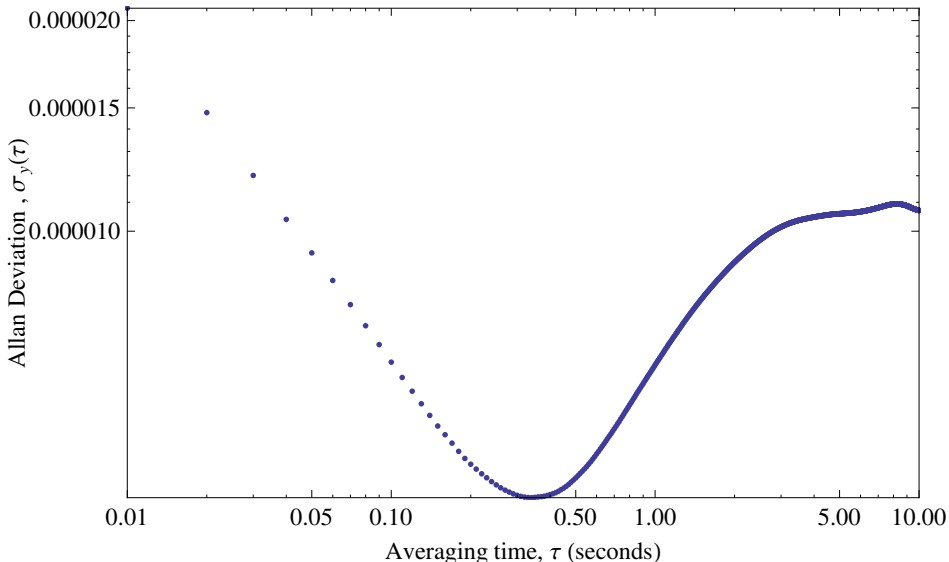

**Figure 17.** Allan deviation of an original equipment manufacturer (OEM) current controller showing shot-noise-limited behavior for $\tau < 0.3$ s. The minimum Allan deviation is $4.2 \times 10^{-6}$ at $\tau = 0.34$ s. The long-term Allan deviation is $10.7 \times 10^{-6}$. This data was acquired with bandwidth filter cut-offs at 0.01 Hz and 10 kHz. The typical rms deviation in the DC current is 0.32 $\mu$A. The current noise density measured with a 10 kHz bandwidth is 0.35 $\mu$A$/\sqrt{\text{Hz}}$.

## 6. Conclusion

We have presented an overview of the best-suited echo interferometric techniques to realize precision measurements of $g$ and $\omega_q$, and we have reviewed coherent transient methods for obtaining the most sensitive measurement of atomic g-factor ratios. The proposed apparatus outlined at the end of Section 3 will combine features of long-lived echo experiments in a non-magnetic chamber in which atoms can be routinely cooled to $\sim 1$ $\mu$K using polarization-gradient cooling with the advantages of vibration stabilization. As a result, it should be possible to load the laser-cooled sample into a deep optical lattice to achieve a several hundred-fold increase in grating contrast and signal size. Echo AIs operating in this setup in an off-resonant configuration using low-cost, high-power lasers appear to be capable of reducing both the statistical and systematic uncertainties to competitive levels. As a result, the relative simplicity of echo AI experiments may be well suited for several precision measurements.

**Acknowledgments:** This work was supported by the Canada Foundation for Innovation, Ontario Innovation Trust, the Natural Sciences and Engineering Research Council of Canada, Ontario Centres of Excellence, the US Army Research Office, and York University. We acknowledge the contributions of Carson Mok, who carried out the gravity experiments, and of Iain Chan, who carried out the magnetic coherence experiments. We thank Tycho Sleator of New York University for the generous loan of Ti:Sapphire laser components.

**Author Contributions:** Brynle Barrett wrote the introductory Sections 1.1 and 1.2, carried out the recoil frequency measurements presented in Section 2, and contributed to the gravity and magnetic coherence experiments presented in Sections 3 and 4, respectively. Adam Carew designed and constructed a modified apparatus for gravity measurements, and he contributed to the measurements of gravitational acceleration, atomic recoil, magnetic coherences, and the design of the auto-locked lasers presented in Section 5. Hermina C. Beica prepared the manuscript, and contributed to the design and characterization of the auto-locked lasers. Andrejs Vorozcovs led the design and construction of the auto-lock controller. Alexander Pouliot constructed and characterized diode laser controllers and carried out an analysis of gravity measurements from the two-pulse AI.

**Conflicts of Interest:** The authors declare no conflict of interest.

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
