# Peer review of "Prospects for Precise Measurements with Echo Atom Interferometry"

_atoms, doi:10.3390/atoms4030019_

Round 1

Author Response

Dear Editor,

We thank you for the supportive comments on our article. We now provide an outline of the changes we have made to the manuscript.

Apart from addressing the referee comments, we have made minor changes to the text of the manuscript and fixed a few additional typos. We have also added three references, and corrected some of the existing references.

In what follows, we mirror each of the comments from the referee in point form and provide our response immediately afterward.

Sincerely,

A. Kumarakrishnan and co-authors.

-          1) Authors should tell us more about the detection technique. How the read-out signal is analyzed to extract the fringe pattern? What is the detection noise, how this noise impacts the statistical uncertainty? ....

Response: We have added the following description of how the signal is processed for recoil measurements near line 255:

“A gated photo-multiplier tube (PMT, 8 x 10^{-5} W/V at 780 nm, noise equivalent power 100 nW) is used to detect the power in the back-scattered field. Figure 3(c) shows an example of the echo signal recorded by the PMT averaged over 16 repetitions of the two-pulse AI. This signal is converted to units of optical power and numerically integrated to obtain a quantity which we term the echo energy. This quantity is proportional to the contrast of the atomic density modulation and the intensity of read-out light incident on the atoms. As a result, the signal is sensitive to both atom number fluctuations and photon number shot noise. This is a drawback compared to fluorescence detection techniques, where the optical transition can be saturated and is therefore less sensitive to photon shot noise [102]. In these experiments, we typically observed a noise floor of 0.1 pJ per shot, or ~ 0.025 pJ after averaging over 16 repetitions, which was dominated primarily by the NEP of the PMT.”

For the gravity measurements, signal processing for the two-pulse AI is described in lines 594-602, and in lines 626-634 for the three-pulse AI.

-          2) To underline the benefit of EAI method for gravity measurement, the authors draw on a comparison with the method described in ref (118). Since 2011, others methods has been investigated (see M. Andia et al., Phys. Rev. A 88, 031605(R) (2013 and R. Charriere et al. Rev. A 85, 013639 (2012)). I think that these references should be at least cited.

Response: The article by Andia et al was already cited in the article. We have added a citation to Charriere et al in sections 1 and 3.1. We have also rewritten section 3.1 to be more consistent with recent work carried out by the cold-atom gravimeter community.

-          3) The authors present succinctly the (low coast) laser source that leads to the required features (in terms of power, spectral tuning range and frequency noise). To my knowledge similar schemes are currently used in many laboratories. Probably I did not get (understand? ) the subtleties. The authors should more highlight the originality of their scheme. The ref 58 should be completed, When I look on the web, using the title of the reference 58, I find a patent?

Response: In lines 870-887, we have pointed out that the novelty of these laser systems is the ability to vacuum-seal the cavity and reduce frequency shifts due to pressure variations. The auto-locking controller uses a variety of feedback algorithms to ensure long-term stability. Such a laser configuration does not seem to be available commercially. We have modified the citation to this work to reflect the fact that the manuscript is in preparation.

Reviewer 2 Report

Dear authors,

This is a well-written manuscript summarizing experiments carried out in the last author's group. The authors describe the challenges of echo atom interferometry. In particular, the experiments to determine gravitational acceleration g and the atomic fine structure constante due to measurement of the atomic recoil frequency.

Some parts of the manuscript are more or less word by word copies of previous paper.  It is up to the editor how to handle that. 

However this is a review paper of previous works and with some reworking of the manuscript and fixing some small problems I support publication of this manuscript. 

Please find below further comments:

14: "exquisite sensitivity"  normal accelerometers provide better performances and are less complex.

page 8  8th line form bottom: brackets are missing the formula 

79: shot-noise limits of < 1 mrad....  please cite 

102: 1600 photon momenta...  please double check that number and citation. Is that a coherent beam splitter?

128: ...appreciable atom loss characteristic of Raman AIs....   comment: Raman AIs are normally not atom shot noise limited. 

128-130: ...Spatial overlap is the only condition required to create an interference pattern,..

 Who would the interference pattern look like if there where just classical beamsplitters? Focusing atoms by micro lenses (standing light wave). Have there been Leggett Garg test?

Formular 18  looks like that there is something wrong!

Figure 4: I don´t get the error bars. They look way to big. 

341: ...that a 100-fold increase.... please cite 

426: ...to apply a a near-resonant....  typo

536, 573: .... all decoherence mechanisms...  could that also be dephasing? please be specific.

Figure10: error bars? It is not obvious that the error bars are so big and the measured/averaged data points line up perfectly next to each other. I think the paper would benefit a lot if you could comment on the error bars.  

706: tip: use a non magnet accelerometer, otherwise you will not be able to distinguish between vibrations and magnetic fields

723: ....is not expected not to affected...   typo?

731: tip:to get the position, one has to integrate 2 times. A seismometer which is sensitive to velocity is maybe a better choice?

778: What means MGE?

Author Response

Dear Editor,

We thank the referee for the supportive comments on our article. We now provide an outline of the changes we have made to the manuscript.

Apart from addressing the referee comments, we have made minor changes to the text of the manuscript and fixed a few additional typos. We have also added three references, and corrected some of the existing references.

In what follows, we mirror each of the comments from the referee in point form and provide our response immediately afterward.

Sincerely,

A. Kumarakrishnan and co-authors.

-          14: "exquisite sensitivity" normal accelerometers provide better performances and are less complex.

Response: It is unclear what the referee means by “normal accelerometers”. Micro-electro-mechanical system (MEMS) accelerometers and capacitive force-balance accelerometers currently dominate the commercial market, and while they are much less complex than a cold-atom sensor, their best performance is typically at the level of 1E-5 g (MEMS) and 1E-7 g (force-balance) due to bias drifts. Falling corner-cube gravimeters, and cold-atom-based gravimeters can achieve absolute measurements of less than 1E-8 g per shot, but are much more complex. Nevertheless, both of these types of instruments are available as portable field devices for high-resolution studies of gravimetry.

-          page 8  8th line form bottom: brackets are missing the formula

Response: We have reviewed this page of the manuscript and we are unable to find the missing brackets. There does not seem to be a problem.

-          79: shot-noise limits of < 1 mrad....  please cite

Response: We have added references to [Le Gouet et al, Appl. Phys. B 92, 133 (2008)] and [Rocco et al, New J. Phys. 14, 093046 (2014)].

-          102: 1600 photon momenta...  please double check that number and citation. Is that a coherent beam splitter?

Response: The number of photon recoil is correct, but the citation we provided [Clade et al Phys. Rev. Lett. 102, 240402 (2009)] was incorrect. We have corrected the citation in the article with a reference to [Cadoret et al, Phys. Rev. Lett. 101, 230801 (2008)], which is an earlier article by the same group. The Bloch-oscillation beam splitter is coherent, in principle, but technical issues related to differential light shifts during the pulse have so far limited the number of photon recoil transfers to ~90 for phase-coherent beam splitters. In the experiment where 1600 photon recoils are transferred, they did not require phase coherence because they were measuring the velocity difference between atomic ensembles using a Ramsey technique.

-          128: ...appreciable atom loss characteristic of Raman AIs....   comment: Raman AIs are normally not atom shot noise limited.

Response: We agree with the referee’s comment that Raman AIs are not typically shot-noise-limited since they do not suffer from low atom numbers (usually N ~ 10^6 atoms participate in the interferometer, yielding a shot-noise limit of ~1/sqrt(N) ~ 1 mrad). However, our intent is to point out that echo AIs require much larger atom numbers (~10^8) to achieve comparable signal-to-noise ratios due to the backscattering detection technique.

-          128-130: ...Spatial overlap is the only condition required to create an interference pattern,…Who would the interference pattern look like if there where just classical beamsplitters? Focusing atoms by micro lenses (standing light wave). Have there been Leggett Garg test?

Response: The quantum mechanical interference pattern obtained from the interference between a momentum states of a single atom is similar to the classical interference pattern one would obtain with a billiard ball model and physical gratings. The difference with the echo AI is that the fringe pattern is a single-atom effect that results from multiple pathway interference arising from a phase grating excitation by standing waves. The fringe pattern can in principle be measured by a phase mask technique, but typically requires many atoms to participate in order to obtain an appreciable signal-to-noise ratio. Although spatial overlap between trajectories is the only condition required to cause interference within a single atom, in order to measure the interference, phase coherence is needed between many atoms in the ensemble. Decoherence mechanisms in an experiment such as spontaneous emission cause the echo AI signal to decrease more severely than if the fringe pattern were classical in nature. Furthermore, the temporal modulation of the echo signal at the two-photon recoil frequency is a purely quantum mechanical effect.

-          Formular 18  looks like that there is something wrong!

Response: We have confirmed that formula 18 is correct.

-          Figure 4: I don´t get the error bars. They look way to big.

Response: All the error bars in Figure 4 are correct. In Fig 4(a), each point is the average of 16 measurements and the error bars represent the standard deviation of the mean of those measurements. In Fig 4(b), the error bars represent the statistical uncertainty for each measurement of the recoil frequency, which was typically 380 ppb. For the entire data set, assuming a white noise distribution, the total statistical uncertainty was 37 ppb.

-          341: ...that a 100-fold increase.... please cite

Response: The work related to the numerical simulations showing a 100-fold increase is presently not yet published, and we would prefer not to cite an unpublished manuscript.

-          426: ...to apply a a near-resonant....  typo

Response: We have corrected this typo.

-          536, 573: .... all decoherence mechanisms...  could that also be dephasing? please be specific.

Response: We use “decoherence” and “dephasing” in the article synonymously, which refers to the loss of phase coherence between atoms participating in the interferometer. However, we have specified that the echo technique cancels only the effect of Doppler dephasing from the velocity distribution of the sample. On both lines 536 and 573, the loss of signal is due to decoherence mechanisms (such as spontaneous emission and magnetic curvature) and a loss of atoms from the interaction zone.

We have modified the sentence at line 536 to read: “The last term in Eq. (22) represents a phenomenological decay, with a time constant $\tau_{\text{decay}}$ that models signal loss due to decoherence mechanisms in the experiment (e.g. from spontaneous emission and the spatial curvature of the ambient magnetic field), as well as the transit time of cold atoms through the interaction zone defined by the excitation beams.”

-          Figure10: error bars? It is not obvious that the error bars are so big and the measured/averaged data points line up perfectly next to each other. I think the paper would benefit a lot if you could comment on the error bars.

Response: In figure 10, the error bars on each point represent the standard deviation of the 16 measurements acquired for each point (as explained in lines 604-609), while the value represented by the points is the normalized average of those 16 measurements. These data indicate that sinusoidal fits are possible despite a significant phase error of 0.7 rad. However, it is true that the error bars do not correctly represent the distribution of the data points, especially near the peaks and troughs of the sinusoid where the value is close to +/-1. The normalization process used on these data tended to skew the points in these regions, but we did not rigorously skew the error bars in the same way. Technically, the error bars should be asymmetric near the peaks and troughs---following a statistical distribution governed by the convolution between an inverse sine function and a Gaussian distribution with the same standard deviation mentioned above. Nevertheless, since the measurement of the gravitational acceleration from these data is mostly determined by the regions where the sinusoid crosses zero, we do not expect this operation to have a strong effect on the measurement.

-          706: tip: use a non magnet accelerometer, otherwise you will not be able to distinguish between vibrations and magnetic fields

Response: We thank the referee for this suggestion.

-          723: ....is not expected not to affected...   typo?

Response: We have corrected this sentence.

-          731: tip:to get the position, one has to integrate 2 times. A seismometer which is sensitive to velocity is maybe a better choice?

Response: We thank the referee for the suggestions.

-          778: What means MGE?

Response: MGE has been defined as “magnetic grating echo” in this sentence.

Round 2

Reviewer 1 Report

Improvements to the paper perfectly respond to questions and precisions that I requested. I am convinced that this paper will be a very useful and a complete reference in the domain of Echo atom interferometry.  It obviously deserves to be published. .

Author Response

Dear Editor,

We are resubmitting the manuscript after addressing suggestions from two referees. One of the referees was fully satisfied with the previous version of the manuscript. The second referee was satisfied with all our responses except the analysis of the data in Figure 10.

Accordingly, we have re-analyzed the data in Figure 10. A better representation of the signal amplitude was calculated by averaging neighbouring data points. This procedure eliminates distortions in the sinusoidal signal amplitude.

Additionally points associated with small signal amplitudes near the zeroes associated with the recoil-modulated signal were removed.

The error bars were estimated on the basis of the probability density function analysis described in Reference 121 (newly introduced). The weighting of the error bars was determined from the signal strength and the probability density function.

Although there is very little change in the overall statistical precision (7 ppm instead of 6 ppm), the fits and the error bars provide a more realistic description of the signal amplitude.

We thank the referee for motivating this analysis.

We hope that the manuscript will be accepted for publication in its revised form.

Sincerely,

A. Kumarakrishnan and co-authors.

Reviewer 2 Report

-          14: "exquisite sensitivity" normal accelerometers provide better performances and are less complex.

Response: It is unclear what the referee means by “normal accelerometers”. Micro-electro-mechanical system (MEMS) accelerometers and capacitive force-balance accelerometers currently dominate the commercial market, and while they are much less complex than a cold-atom sensor, their best performance is typically at the level of 1E-5 g (MEMS) and 1E-7 g (force-balance) due to bias drifts. Falling corner-cube gravimeters, and cold-atom-based gravimeters can achieve absolute measurements of less than 1E-8 g per shot, but are much more complex. Nevertheless, both of these types of instruments are available as portable field devices for high-resolution studies of gravimetry.

--)It is for me misleading to talk about the exquisite sensitivity of atom interferometry. To reach with cold atoms a level of 1e-7g/Hz^0.5 it takes years of work and to buy an absolute cold-atom-based gravitmeter costs some 100k$. Middlemiss, R. P. et al. Nature 531, 614–617 (2016) claims to reach a sensitivity of ~4E-8g/Hz^0.5 with MEMS. However, I agree that there is an advantage for “absolute measurements” with cold atoms.

-          page 8  8th line form bottom: brackets are missing the formula

Response: We have reviewed this page of the manuscript and we are unable to find the missing brackets. There does not seem to be a problem.

--)I am sorry it is page 2:   8th line form bottom: brackets are missing the formula

-          79: shot-noise limits of < 1 mrad....  please cite

Response: We have added references to [Le Gouet et al, Appl. Phys. B 92, 133 (2008)] and [Rocco et al, New J. Phys. 14, 093046 (2014)].

-          102: 1600 photon momenta...  please double check that number and citation. Is that a coherent beam splitter?

Response: The number of photon recoil is correct, but the citation we provided [Clade et al Phys. Rev. Lett. 102, 240402 (2009)] was incorrect. We have corrected the citation in the article with a reference to [Cadoret et al, Phys. Rev. Lett. 101, 230801 (2008)], which is an earlier article by the same group. The Bloch-oscillation beam splitter is coherent, in principle, but technical issues related to differential light shifts during the pulse have so far limited the number of photon recoil transfers to ~90 for phase-coherent beam splitters. In the experiment where 1600 photon recoils are transferred, they did not require phase coherence because they were measuring the velocity difference between atomic ensembles using a Ramsey technique.

--) Thank you.

-          128: ...appreciable atom loss characteristic of Raman AIs....   comment: Raman AIs are normally not atom shot noise limited.

Response: We agree with the referee’s comment that Raman AIs are not typically shot-noise-limited since they do not suffer from low atom numbers (usually N ~ 10^6 atoms participate in the interferometer, yielding a shot-noise limit of ~1/sqrt(N) ~ 1 mrad). However, our intent is to point out that echo AIs require much larger atom numbers (~10^8) to achieve comparable signal-to-noise ratios due to the backscattering detection technique.

--) Perfect!

-          128-130: ...Spatial overlap is the only condition required to create an interference pattern,…Who would the interference pattern look like if there where just classical beamsplitters? Focusing atoms by micro lenses (standing light wave). Have there been Leggett Garg test?

Response: The quantum mechanical interference pattern obtained from the interference between a momentum states of a single atom is similar to the classical interference pattern one would obtain with a billiard ball model and physical gratings. The difference with the echo AI is that the fringe pattern is a single-atom effect that results from multiple pathway interference arising from a phase grating excitation by standing waves. The fringe pattern can in principle be measured by a phase mask technique, but typically requires many atoms to participate in order to obtain an appreciable signal-to-noise ratio. Although spatial overlap between trajectories is the only condition required to cause interference within a single atom, in order to measure the interference, phase coherence is needed between many atoms in the ensemble. Decoherence mechanisms in an experiment such as spontaneous emission cause the echo AI signal to decrease more severely than if the fringe pattern were classical in nature. Furthermore, the temporal modulation of the echo signal at the two-photon recoil frequency is a purely quantum mechanical effect.

--)Thank you. 

-          Formular 18  looks like that there is something wrong!

Response: We have confirmed that formula 18 is correct.

-          Figure 4: I don´t get the error bars. They look way to big.

Response: All the error bars in Figure 4 are correct. In Fig 4(a), each point is the average of 16 measurements and the error bars represent the standard deviation of the mean of those measurements. In Fig 4(b), the error bars represent the statistical uncertainty for each measurement of the recoil frequency, which was typically 380 ppb. For the entire data set, assuming a white noise distribution, the total statistical uncertainty was 37 ppb.

-          341: ...that a 100-fold increase.... please cite

Response: The work related to the numerical simulations showing a 100-fold increase is presently not yet published, and we would prefer not to cite an unpublished manuscript.

-          426: ...to apply a a near-resonant....  typo

Response: We have corrected this typo.

-          536, 573: .... all decoherence mechanisms...  could that also be dephasing? please be specific.

Response: We use “decoherence” and “dephasing” in the article synonymously, which refers to the loss of phase coherence between atoms participating in the interferometer. However, we have specified that the echo technique cancels only the effect of Doppler dephasing from the velocity distribution of the sample. On both lines 536 and 573, the loss of signal is due to decoherence mechanisms (such as spontaneous emission and magnetic curvature) and a loss of atoms from the interaction zone.

We have modified the sentence at line 536 to read: “The last term in Eq. (22) represents a phenomenological decay, with a time constant $\tau_{\text{decay}}$ that models signal loss due to decoherence mechanisms in the experiment (e.g. from spontaneous emission and the spatial curvature of the ambient magnetic field), as well as the transit time of cold atoms through the interaction zone defined by the excitation beams.”

--)Perfect!

-          Figure10: error bars? It is not obvious that the error bars are so big and the measured/averaged data points line up perfectly next to each other. I think the paper would benefit a lot if you could comment on the error bars.

Response: In figure 10, the error bars on each point represent the standard deviation of the 16 measurements acquired for each point (as explained in lines 604-609), while the value represented by the points is the normalized average of those 16 measurements. These data indicate that sinusoidal fits are possible despite a significant phase error of 0.7 rad. However, it is true that the error bars do not correctly represent the distribution of the data points, especially near the peaks and troughs of the sinusoid where the value is close to +/-1. The normalization process used on these data tended to skew the points in these regions, but we did not rigorously skew the error bars in the same way. Technically, the error bars should be asymmetric near the peaks and troughs---following a statistical distribution governed by the convolution between an inverse sine function and a Gaussian distribution with the same standard deviation mentioned above. Nevertheless, since the measurement of the gravitational acceleration from these data is mostly determined by the regions where the sinusoid crosses zero, we do not expect this operation to have a strong effect on the measurement.

--)The behavior of the statistics is for me highly unexpected. Even shrinking the error bar by a factor of 4 (in order to get the error bar for the averaged value) doesn´t explain that the points (first figure) line up that perfectly close to +/-1.The scatter around a smooth curve is too small to be consistent with the error bars shown.
The scatter in this area is clearly smaller than the size of the blue circle. According to a Gaussian distribution a ~third of the points should deviate by at least one standard deviation, which is in this case one quarter (=1/(16)^0.5) of the error bar shown. Apparently this mismatch between the error bar size and the points scatter is a data processing artefact.The paper would benefit a lot if this discrepancy would be removed, especially the error analysis is a central part of the sensitivity estimate of the paper.  

-          706: tip: use a non magnet accelerometer, otherwise you will not be able to distinguish between vibrations and magnetic fields

Response: We thank the referee for this suggestion.

-          723: ....is not expected not to affected...   typo?

Response: We have corrected this sentence.

-          731: tip:to get the position, one has to integrate 2 times. A seismometer which is sensitive to velocity is maybe a better choice?

Response: We thank the referee for the suggestions.

-          778: What means MGE?

Response: MGE has been defined as “magnetic grating echo” in this sentence.

Author Response

(The authors gave the same response as above.)
